# Modulation of KDM1A with vafidemstat rescues memory deficit and behavioral alterations

Tamara Maes[1]*, Cristina Mascaró[1], David Rotllant[1], Michele Matteo Pio Lufino[1], Angels Estiarte[1¤a], Nathalie Guibourt[1], Fernando Cavalcanti[1], Christian Griñan-Ferré[2], Mercè Pallàs[2], Roser Nadal[3], Antonio Armario[3], Isidro Ferrer[4], Alberto Ortega[1], Nuria Valls[1¤b], Matthew Fyfe[1¤c], Marc Martinell[1¤d], Julio César Castro Palomino[1¤e], Carlos Buesa Arjol[1]

**1** Oryzon Genomics, S.A., Cornellà de Llobregat, Spain, **2** Faculty of Pharmacy and Food Sciences, Institute of Neuroscience, University of Barcelona, Barcelona, Spain, **3** Institut de Neurociències, Universitat Autònoma de Barcelona, Bellaterra, Spain, **4** Institut de Neuropatologia, Servei Anatomia Patologica, IDIBELL-Hospital Universitari de Bellvitge, L'Hospitalet de Llobregat, Spain

¤a Current Address: Verseon, Fremont, California, United States of America
¤b Current Address: Alba-Cells, Cerdanyola del Vallès, Barcelona, Spain
¤c Current Address: Sitryx therapeutics, Oxford, United Kingdom
¤d Current Address: Minoryx, Mataró, Barcelona, Spain
¤e Current Address: Palobiofarma, Mataró, Barcelona, Spain
* tmaes@oryzon.com

**Data Availability Statement:** Data are available on Mendeley (DOI: https://data.mendeley.com/datasets/sc5tfwt4nr/1). Genome wide expression data are available in NCBI (GSE100413).

## Abstract

Transcription disequilibria are characteristic of many neurodegenerative diseases. The activity-evoked transcription of immediate early genes (IEGs), important for neuronal plasticity, memory and behavior, is altered in CNS diseases and governed by epigenetic modulation. KDM1A, a histone 3 lysine 4 demethylase that forms part of transcription regulation complexes, has been implicated in the control of IEG transcription. Here we report the development of vafidemstat (ORY-2001), a brain penetrant inhibitor of KDM1A and MAOB. ORY-2001 efficiently inhibits brain KDM1A at doses suitable for long term treatment, and corrects memory deficit as assessed in the novel object recognition testing in the Senescence Accelerated Mouse Prone 8 (SAMP8) model for accelerated aging and Alzheimer's disease. Comparison with a selective KDM1A or MAOB inhibitor reveals that KDM1A inhibition is key for efficacy. ORY-2001 further corrects behavior alterations including aggression and social interaction deficits in SAMP8 mice and social avoidance in the rat rearing isolation model. ORY-2001 increases the responsiveness of IEGs, induces genes required for cognitive function and reduces a neuroinflammatory signature in SAMP8 mice. Multiple genes modulated by ORY-2001 are differentially expressed in Late Onset Alzheimer's Disease. Most strikingly, the amplifier of inflammation S100A9 is highly expressed in LOAD and in the hippocampus of SAMP8 mice, and down-regulated by ORY-2001. ORY-2001 is currently in multiple Phase IIa studies.

**Funding:** None of the authors was a direct beneficiary but the studies were funded by Oryzon Genomics S.A. (https://www.oryzon.com) and co-funded by grants or loans to [1] Oryzon Genomics S.A., Cornellá de Llobregat, Spain or [2] Fundació Bosch i Gimpera, Universitat de Barcelona, Barcelona, Spain or [3] Institut de Neurociències, Universitat Autònoma de Barcelona, Bellaterra, Spain: [1] CEN-20081013 and CEN-20101023 of the CENIT program of CDTI, Spanish Ministry of Industry, Tourism and Commerce (https://www.cdti.es/); by grant [1] RD08-2-0014 of CIDEM, Generalitat de Catalunya (https://www.accio.gencat.cat/); by grants [1] 20100902VEN and 20150202 of the Alzheimer's Drug Discovery Foundation (www.alzdiscovery.org); by [1] FP7 grant 278871 of the European Union (https://ec.europa.eu/research/fp7/index_en.cfm); and by [1, 2, 3] RTC-2015-3898-1 and [1, 2] RTC-2016-4955-1 of the RETOS program of CDTI, Spanish Ministry of Science, Innovation and Universities (https://www.cdti.es/).

**Competing interests:** I have read the journal's policy and the authors of this manuscript have the following competing interests: T.M. and C.B. are founders, executive directors and shareholders of Oryzon Genomics S.A.; C.M., D.R., M.M.P.L., AO, J.S. and E.C. are employees; and A.E., N.G., N.V., M.F., M.M. and J.C.C.P. are former employees of Oryzon Genomics S.A.; I.F. is former member of the SAB of Oryzon Genomics S.A. A.O., M.F., M.M., A.E., N.V, J.C.C.P., T.M., C.M., D.R., C.G.F, M.P., R.N., A.A., are listed as inventors on one or several of the following patent applications of Oryzon Genomics S.A. related to this work: WO2012/013728; WO2013/057320; WO2016/198649; WO2017/158136; WO2019/025588. The authors have no other relevant affiliations or financial involvement with any organization or entity with a financial interest in or financial conflict with the subject matter or materials discussed in the manuscript apart from those disclosed. This does not alter our adherence to PLOS ONE policies on sharing data and materials with the following exceptions: Polyphemous, proprietary software of Oryzon Genomics S.A., is not available as an exportable .exe code.

## Introduction

KDM1A is a flavin adenine dinucleotide (FAD) dependent amine oxidase that acts primarily as a histone demethylase [1]. KDM1A is recruited by zinc finger (ZNF) TFs into repressive complexes, interacts tightly with RCOR and HDAC1/2 [2] and demethylates primarily H3K4me2/1 marks, associated with the active transcription state [3]. Nevertheless, KDM1A was also identified in protein complexes associated with activation of transcription, and in this context the protein was described to demethylate H3K9me2/1 [4] or H4K20me2 [5].

KDM1A is implicated in a variety of biological processes. Required during embryogenesis and normal hematopoiesis, it is over-expressed in certain cancer types and plays a key role in leukomogenesis. Certain types of leukemia and solid cancer types like small cell lung cancer are highly sensitive to both KDM1A knock-down and inhibition [6–8], and several selective KDM1A inhibitors are in Phase I and II clinical trials for the treatment of hematological or solid tumor indications.

KDM1A is also required for neurogenesis, and has a role in neuron progenitor cells (NPCs) proliferation [9–11] and terminal differentiation [12–14]. While KDM1A-mediated repression of gene expression (GE) is required for NPC maintenance, the morphogenic properties of KDM1A require dimming of the repressive activity of KDM1A. This is achieved through: rapid feedback mechanisms to lower KDM1A expression level described in olfactory sensor neurons, accelerated protein degradation through PHF15/JADE-2, or alternative splicing [15–17]. KDM1A is expressed in a ubiquitous fashion throughout the body, yet the brain and more specifically, neurons, express developmentally regulated splice forms that incorporate a small additional exon (E8a = DTVK). While there is no consensus on its exact mechanism of action, there is a general agreement that the neuron specific splice form counteracts the ubiquitous splice form and is required for neuronal differentiation and memory. This idea is supported both by the neuro splice form selective KDM1A knockouts and by isoform usage during neuronal differentiation [5, 17–18].

Here, we describe the development and pharmacological characterization of the brain penetrant KDM1A inhibitor ORY-2001 (vafidemstat), reveal its capacity to rescue cognitive deficits and behavior alterations in rodent models, and report on the compound's mechanism of action.

## Materials and methods

The following information is provided in S2 File: Supporting Methods

- Contact for reagents and resource sharing

- Experimental models and samples

  Cell Lines, Human Brain Samples, Human CSF samples, Mouse Pharmacokinetics, Rat Pharmacokinetics, Rat Pharmacodynamics, Mouse MPTP neurotoxicity model, Mouse PEA behavior model, Mouse MAOB inhibition, Rat tyramine pressure response model, Mouse L-5-HTP model, Mouse MAOA inhibition, SAMR1 and SAMP8 model, Rat isolation rearing model.

- Method details

  Biochemistry and Biophysics: Biochemical Assays, KDM1A Splice Form Kinetics Analysis, KDM1A Binding Analysis, MALDI-TOF Mass. Spectrometry.

  Cell biology: THP-1 FACS Analysis.

  Pharmacokinetic and Pharmacodynamic Analyses: Hematology, *Ex vivo* analysis of KDM1A Target Engagement, MPTP Toxicity Test, PEA-induced Symptoms Test, *Ex vivo*

analysis of MAO-B Target Inhibition, Tyramine Pressure Response Test, L5-HTP-induced Symptoms Test, *Ex vivo* analysis of MAO-A Target Inhibition.

Efficacy Studies: Novel Object Recognition Test, Open Field, Elevated Plus Maze, Resident Intruder, Three Chamber Test.

Gene expression analysis: Oligo Design, Microarray Fabrication and QC, RNA extraction, Microarray hybridization and analysis, qRT-PCR

Protein expression and interaction analysis: Western Blot, Chemoproteomics, Chemoprobe ELISAs, KDM1A interaction ELISAs, Double labeling immunofluorescence, ELISA.

- Quantification and statistics

- Data deposition and software

- Additional resources

## Results

### ORY-2001 is a covalent inhibitor of KDM1A and MAO-B

Computational models based on X-ray structures of KDM1A, MAO-A and MAO-B [19–22], fine-tuned with preliminary SAR data, were used to design and synthesize >800 KDM1A inhibitors using the monoamine oxidase (MAO) inhibitor tranylcypromine (TCP) [23,24] as a chemical starting point. Molecules were ranked for KDM1A, MAO-A and MAO-B inhibitory activity using biochemical assays as described in Methods, and represented in a bubble diagram (S1A Fig). Fruit of these efforts, we identified ORY-2001 (Fig 1A), a compound with mean $IC_{50}$ values of 101 ± 40 nM in KDM1A and 73 ± 34 nM in MAO-B (Fig 1B and 1C). KDM1A is inhibited by covalent binding of ORY-2001 to the FAD cofactor, as shown by changes in the absorbance spectrum of the FAD cofactor and by MALDI-TOF Mass Spectrometry (S1B and S1C Fig). The FAD adducts are formed by cyclopropyl ring opening; similarly to adducts formed with TCP described in the literature [20]. ORY-2001 does not bind free FAD and displays excellent selectivity for inhibition of KDM1A and MAO-B over other FAD containing monoamine oxidases (S1A Table). The *in vitro* kinetics of inhibition of KDM1A; or of the KDM1A, KDM1A-E8a or KDM1A-E2a/8a isoforms in complex with RCOR1, were analyzed using a H3K4me2 tail peptide as a substrate. The $k_{inact}/K_i$ value for ORY-2001 using the pure recombinant enzyme was 14509 $M^{-1}sec^{-1}$ (S1 Table). Addition of RCOR1 increased the $k_{inact}/K_i$ values, but all splice forms were efficiently inhibited by ORY-2001 (S1 Table). KDM1A inhibitors have been described to induce differentiation in acute myloid leukemia (AML) cell lines. The cellular activity of ORY-2001 was evaluated using a fluorescent assisted cell sorting (FACS) based CD11B induction assay in the AML cell line THP-1 [8], and yielded an $EC_{50}$ of 21 nM (Fig 1D). A chemoprobe based assay [25, 26] was used to directly assess KDM1A target engagement in the same cell line; and the $EC_{50}$ was ~ 1 nM (Fig 1E).

The specificity of ORY-2001 was evaluated further in a panel of 19 epigenetic targets. At 10 μM, none of the epigenetic targets were >15% inhibited (S1B Table). In a broader panel, 86 additional pharmaceutical targets (CEREP) were used to test ORY-2001 at a concentration of 10 μM and only two targets showed >50% inhibition of enzymatic activity or ligand binding (S1C Table): MAO-B (94%) and the I2 imidazoline receptor (71%), respectively. MAO-A inhibition at 10 μM was 37%, in line with internal data. Altogether, these data illustrate that ORY-2001 is selectively inhibiting KDM1A and MAO-B *in vitro*.

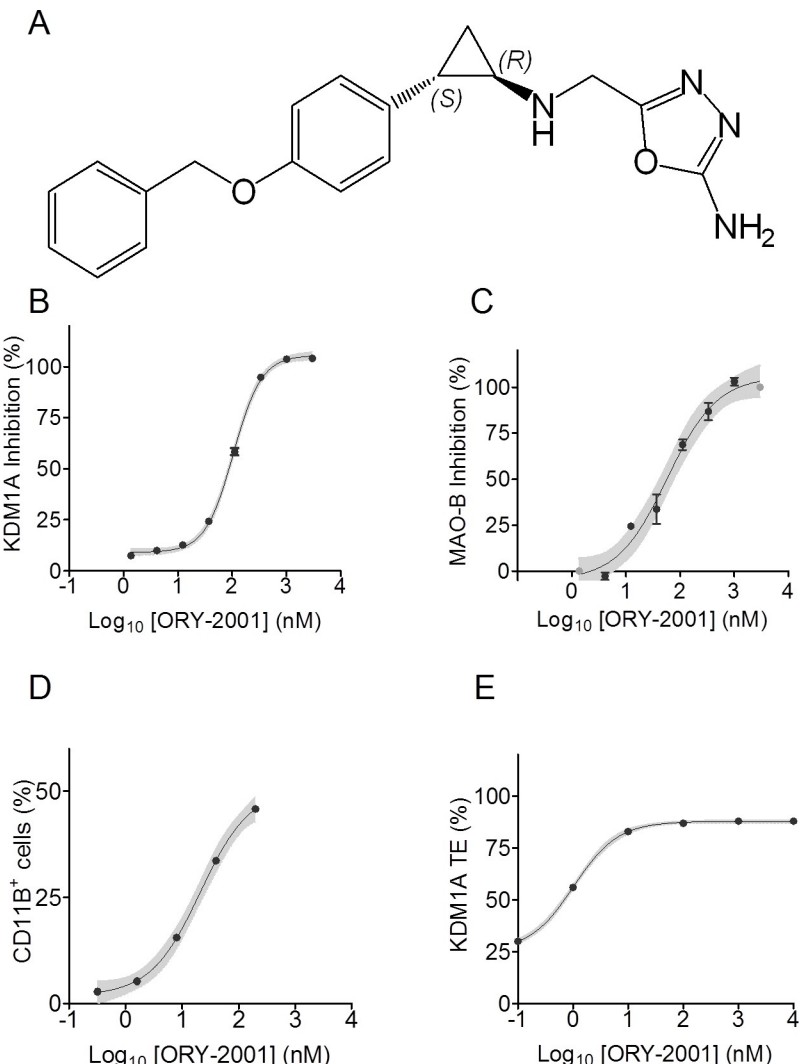

**Fig 1. ORY-2001 is a covalent inhibitor of KDM1A and MAO-B.** (A) Chemical structure of ORY-2001. (B) A representative dose-response curve of rKDM1A inhibition by ORY-2001 ($IC_{50}$ KDM1A = 105 nM); N = 1, n = 2, mean ± SD. (C) A representative dose-response curve of MAO-B inhibition by ORY-2001 ($IC_{50}$ MAO-B = 58 nM); N = 1, n = 2, mean ± SD. (D) Dose curve of induction of the differentiation marker CD11B in THP-1 cells by ORY-2001 ($EC_{50}$ = 21 nM); N = 1, n = 1; (E) Dose curve of KDM1A target engagement in THP-1 cells treated 24 hr with ORY-2001 ($EC_{50}$ = 1 nM); N = 1, n = 3, mean ± SD.

The pharmacokinetics, platelet pharmacodynamics and KDM1A target engagement and selectivity relative to MAO-B and MAO-A, were characterized in rodents and can be found in the Supporting Materials (S2 Table, S2 Fig and S1 File). Summarized, ORY-2001 efficiently crosses the BBB in rodents and inhibits KDM1A > MAO-B > MAO-A *in vivo*.

ORY-2001 was tested in maximum tolerated dose (MTD) studies. As expected, the dose limiting toxicity of ORY-2001 in 5 day MTD studies was the hematopoietic impact associated with KDM1A inhibition, as reflected by a reversible and dose dependent impact on platelets (PLTs) (S2 Fig). The MTD studies and pharmacokinetics data served to estimate doses for long term treatment in efficacy models.

## ORY-2001 rescues the memory of SAMP8 mice

The SAMP8 mouse is a non-transgenic model for accelerated aging and Alzheimer's disease. These animals have a shortened life-span and accumulate many of the histopathological, cognitive and behavioral hallmarks present in the human AD pathology [27, 28]. SAMP8 mice have been characterized by genome wide expression analysis and deep sequencing, and mutations contributing to the phenotype have been identified [29, 30]. The evolution of the memory deficit of SAMP8 mice (Fig 2A) can be reliably assessed using the novel object recognition test (NORT) and is significant in both sexes from 5 months of age. The relative independence of the NORT parameters from potential interfering factors (stress, physical condition,. . .) as well as the reproducibility of the phenotype and the ample test window provide the NORT assay with optimal characteristics for pharmacological dose range testing. ORY-2001 was tested in a series of experiments in SAMP8 mice (S3 Table) summarized below. Vehicle treated SAMR1 mice were included as a reference strain.

**Therapeutic dose range finding.** ORY-2001 was administered for 2 to 4 months in drinking water, at doses ranging from 0.11 to 3.2 mg/kg/day in female or male SAMP8 mice (Exp1 to Exp3). Treatment was well tolerated. At the highest dose a mild weight loss and toll on fitness was observed, accompanied by a ~ 50% reduction of PLT level, indicating that the maximum tolerated dose for long term treatment had been reached. Vehicle treated SAMP8 mice were unable to discriminate novel from familiar objects, and this deficit was dose dependently restored by treatment (Fig 2B and S3A–S3D Fig). The lowest dose (0.11 mg/kg/day) provided ~ 50% rescue. The therapeutic window was estimated to be at least 30 fold. KDM1A target engagement in samples from the cortex of animals treated with ORY-2001 was assessed using chemoprobe ELISA [25, 26] and correlated with the observed functional effects on memory as assessed by NORT (Fig 2C).

**KDM1A inhibition is key to the memory rescue by ORY-2001.** To evaluate the relative contribution of the KDM1A and MAO-B inhibition, we tested the brain-penetrant selective KDM1A inhibitor ORY-LSD1 (Exp3) and the selective MAO-B inhibitor rasagiline (RSG,

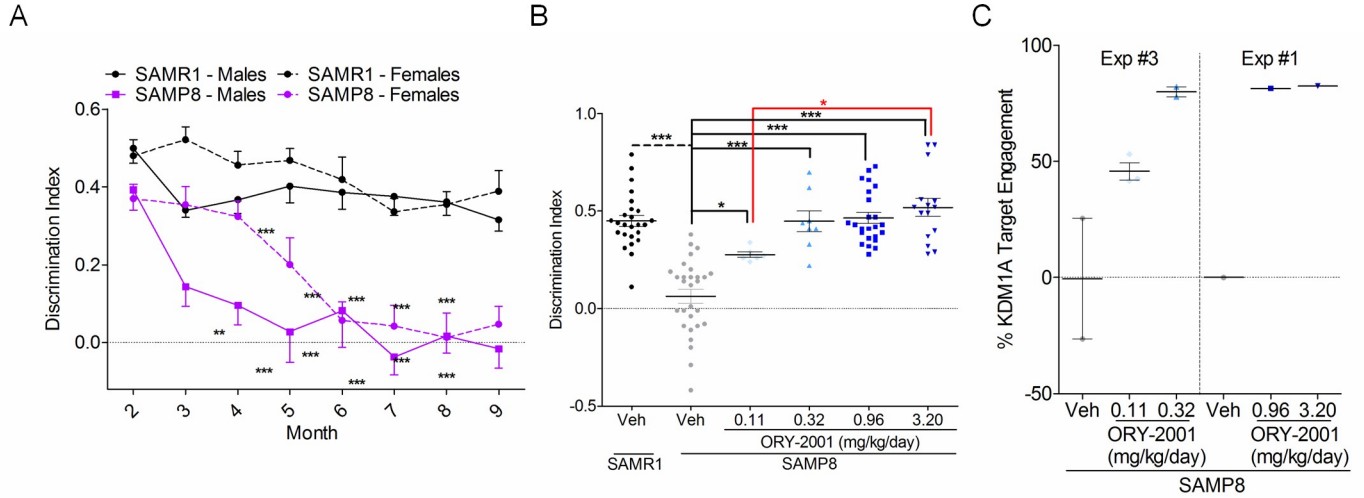

**Fig 2. ORY-2001 rescues the memory deficit in SAMP8 mice.** (A) Novel Object Recognition Test (NORT) Discrimination Index (DI) meta-analysis of untreated animals showing cognitive decline in SAMP8 vs. SAMR1 mice (Males $F_{7,224}$ = 1.970; p = 0.060 Interaction; Females $F_{7,224}$ = 2.403; p = 0.022 Interaction; n = 15 group/age). (B) ORY-2001 dose response effect on DI in the NORT ($F_{4,77}$ = 24.74; p < 0.0001; SAMR1 veh N = 25; SAMP8 veh N = 29, ORY-2001 0.11 mg/kg/day N = 6, 0.32 mg/kg/day N = 8, 0.96 mg/kg/day N = 24, 3.20 mg/kg/day N = 15). (C) Dose response of KDM1A target engagement in the brain cortex of SAMP8 mice at end of 2 month treatment with ORY-2001.

Exp4) in parallel with ORY-2001 in the SAMP8 model. The target selectivity profile of ORY-LSD1 is provided in S1 Table and its PK profile in S2 Table. The doses of ORY-2001 and ORY-LSD1 were normalized to the equivalent impact on hematopoiesis found in MTD experiments. RSG was used at 3 mg/kg/day, a dose sufficient to achieve complete MAO-B inhibition [31]. As in previous experiments, ORY-2001 (0.32 and 0.96 mg/kg/day) completely restored the discrimination capacity. ORY-LSD1 (0.1 and 0.3 mg/kg/day) was also efficacious although less than ORY-2001 (Fig 3A). RSG demonstrated a tendency for improvement without reaching significance (p = 0.12) (Fig 3B). These results show that KDM1A inhibition, the main pharmacological activity of ORY-2001 *in vivo* as assessed in the *ex vivo* target engagement or inhibition assays, is key to memory rescue in this model, although MAO-B inhibition could provide a minor contribution.

**ORY-2001 has symptomatic and disease-modifying potential.** We then evaluated how the treatment window affects the therapeutic effect of ORY-2001. A short 5 day treatment at 0.96 mg/kg/day (Exp5, Fig 4A) was equally effective as a one month treatment (Exp6, Fig 4B and Exp8, S3 Table) to rescue the discrimination capacity in the NORT. This fast response may point at a symptomatic effect.

ORY-2001 was tested in a complete two-period cross-over experiment (Exp6). Five month old female SAMP8 mice were divided in 4 groups, treated for either 2 months with 0.32 mg/kg/day ORY-2001 (complete period), for 1 month with ORY-2001 and 1 month with vehicle (withdrawal), for 1 month with vehicle and 1 month with ORY-2001 (delayed start) or treated with vehicle for 2 months (vehicle). All animals were tested by NORT at the end of treatment (month 7). The discrimination capacity was restored in the complete period and delayed start group. Interestingly, the withdrawal group remained significantly improved relative to the vehicle treated group 1 month after treatment interruption (Fig 4B). Brain pharmacokinetics and pharmacodynamics (S2A Table, S2B Fig) show that brain compound and target engagement levels return to baseline in less than 1 and ~ 3 days, respectively. The therapeutic effect

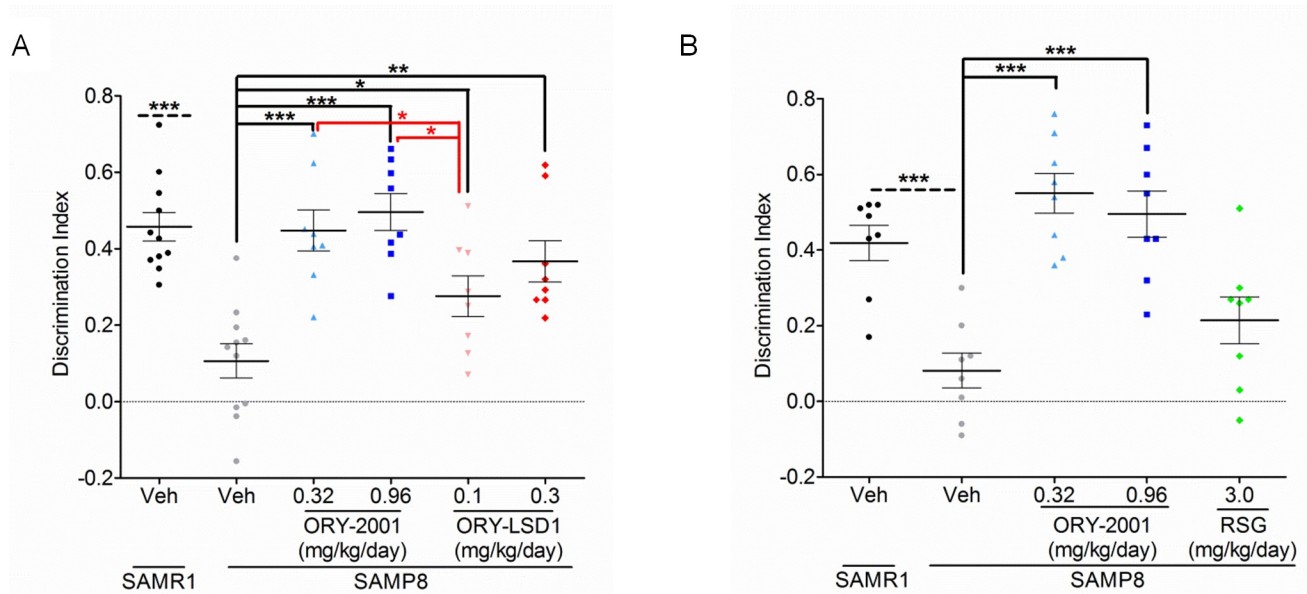

**Fig 3. KDM1A inhibition is key for memory rescue by ORY-2001.** (A) Comparison of the effect of ORY-2001 and ORY-LSD1 on the DI in the NORT reflects higher efficacy of ORY-2001 in SAMP8 mice ($F_{4,38}$ = 10.40; p < 0.0001; Vehicle: N = 11/group; ORY-2001 and ORY-LSD1: N = 8/dose). (B) Comparison of the effect of ORY-2001 and RSG on the DI in the NORT reflects higher efficacy of ORY-2001 in SAMP8 mice ($F_{3,28}$ = 16.84; p < 0.0001; N = 8/group).

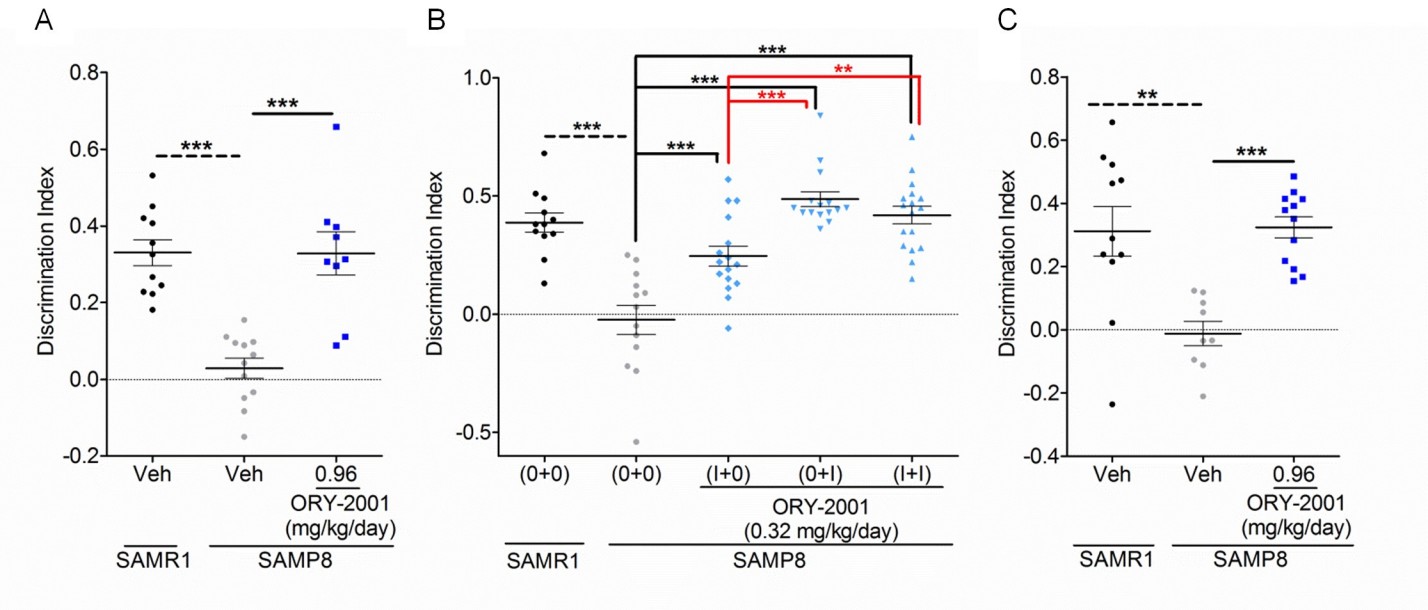

**Fig 4. Symptomatic and disease modifying potential of ORY-2001.** (A) Effect of one week ORY-2001 treatment on NORT DI (t(19) = 5.230; p < 0.0001; N = 9-12/group), assessed 2 hr after training (B) Effect of ORY-2001 on NORT DI in a crossover design; 0: one month treatment with vehicle; I: one month treatment with ORY-2001 ($F_{3,57}$ = 25.65; p < 0.0001; N = 12-17/group). (C) Effect of ORY-2001 on DI in the NORT of aged SAMP8 mice (t(19) = 6.600; p < 0.0001; N = 9-12/group). Means and SEM are represented. SAMR1 and SAMP8 vehicle groups were compared by Student t-test. Within SAMP8 mice, different drug treatments were compared by t-test or by oneway-ANOVA with Dunnett and SNK post-hoc analysis. *p < 0.05, **p < 0.01, ***p < 0.001.

therefore outlasts the target turnover, and provide the first indication that ORY-2001 may have disease-modifying potential.

Finally, the memory deficit was assessed in aged SAMP8 mice. Eight month old SAMP8 mice were treated for 4 months with ORY-2001 or vehicle, and compared with vehicle treated SAMR1 mice of the same age (Exp7). The NORT of the SAMR1 mice indicated that cognitive decline had also started in the reference strain (mean DI = 0.3), but again, treatment fully rescued the performance of the SAMP8 mice (Fig 4C).

**ORY-2001 modulates the hippocampal GE profile.** Changes underlying the memory rescue in the NORT, were analyzed using a microarray-based survey of pooled hippocampal samples from SAMR1 mice and vehicle or ORY-2001 treated SAMP8 mice. To maximize the probability of identifying changes, the highest dose experiment was used (Exp1), and to assess both the differential expression between genotype or caused by treatment, all samples were compared to the vehicle treated SAMP8 mice using two color hybridization.

The survey confirmed a large part of the previously described genotype-associated differences in hippocampal expression in SAMP8 versus SAMR1 mice [29], including higher expression of a set of genes in a region of chr4 amplified in SAMP8 mice including *Ccl19*, *Ccl27*, *Oprs1* and higher expression of *Kif5b* and *Ppp1r1a* in SAMR1, which served for genotype confirmation (Fig 5A). Strikingly, a part of the genes expressed at higher levels in SAMP8 vs SAMR1 mice, were down-regulated by treatment with ORY-2001. These genes included: *Mela*, *Iap*, *Agxt2l1* and genes involved in (neuro-)inflammation including *S100a9*, T-cell receptor beta genes, Twist, *Cd3d, and Ly6c*. Genes found up-regulated by treatment in the SAMP8 hippocampus included *Baiap3*, a protein involved in retrograde trafficking [32]; *Prph*, a gene affected by mutations in Amyotrophic Lateral Sclerosis [33]; *Fabp7* [34,35] and *Doc2a* [36], genes required for cognitive function and memory; *Kremen2* and *Rspo1*, regulators of the WNT pathway [37]. Up-regulated mildly by treatment, but interesting collectively, were

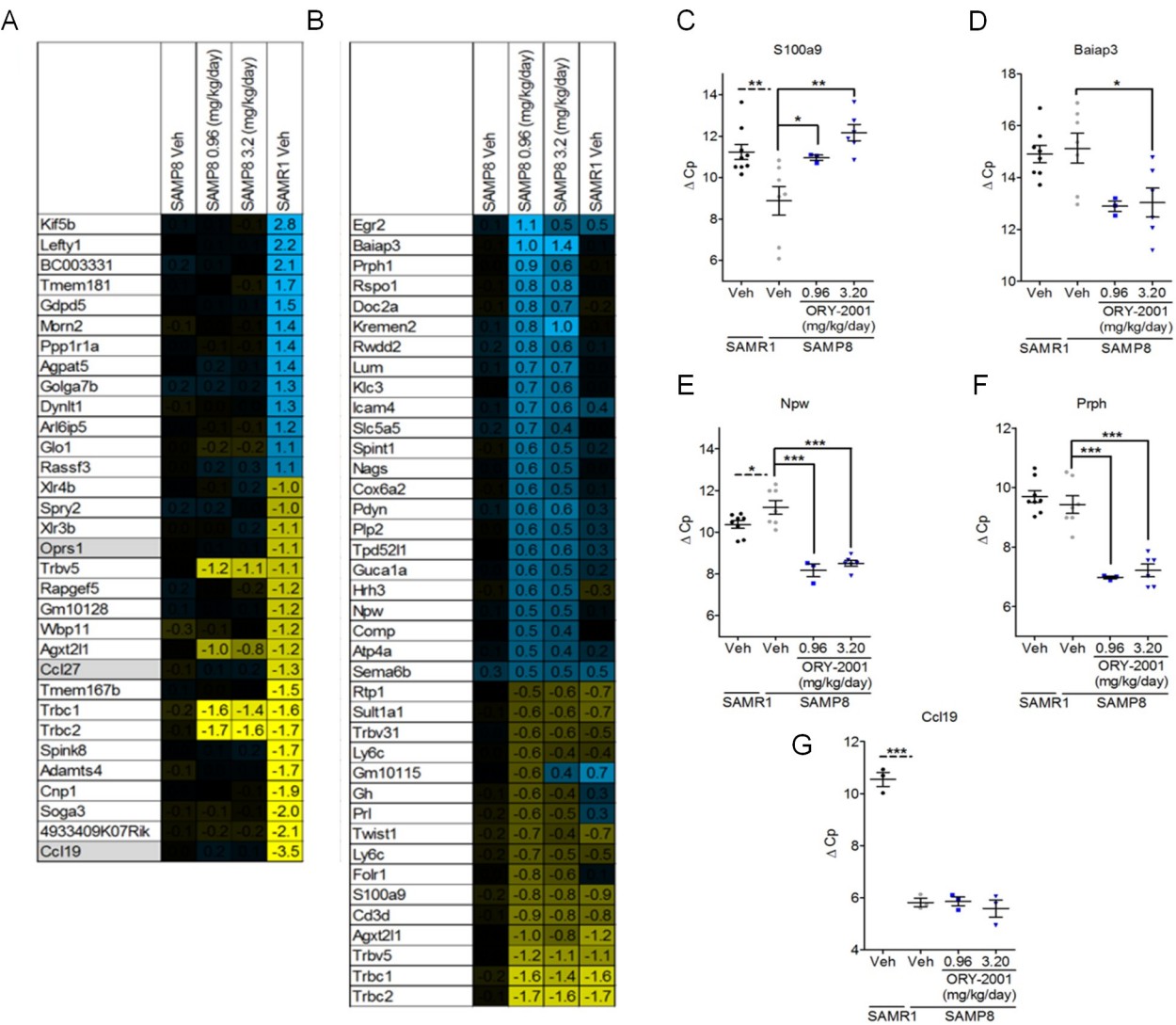

**Fig 5. ORY-2001 induces GE changes in the hippocampus of SAMP8 mice.** Microarray analysis of hippocampal tissue from vehicle- and ORY-2001-treated SAM mice: (A) GE changes in SAMR1 (genes with changes > 2-fold are shown) compared to vehicle-treated SAMP8 mice. Genes are ranked according to the highest expression change. Genes on chr4 with higher copy number in SAMP8 compared to SAMR1 mice (*Oprs1*, *Ccl27* and *Ccl19*) are genotype controls and highlighted in grey. (B) GE changes in SAMP8 mice treated with ORY-2001 compared to vehicle-treated SAMP8 mice (genes with changes > 1.4-fold in ORY-2001 0.96 mg/kg/day samples and > 1.32-fold in ORY-2001 3.2 mg/kg/day samples) are shown. Genes are ranked according to the highest expression change in the SAMP8 0.96 mg/kg/day condition. All values represent Log$_2$(Fold Change). The survey was performed on pooled samples from each group. Veh = Vehicle. qRT-PCR analysis of the effect of ORY-2001 treatment on hippocampal mRNA expression: (C) *S100a9* (F$_{2,13}$ = 9.224; p = 0.0032), (D) *Baiap3* (F$_{2,13}$ = 4.933; p = 0.025), (E) *Npw* (F$_{2,13}$ = 34.66; p < 0.0001), (F) *Prph* F$_{2,13}$ = 26.47; p < 0.0001) (SAMR1 veh N = 8; SAMP8 veh N = 7, ORY-2001 0.96 mg/kg/day N = 3, 3.20 mg/kg/day N = 6) and (G) *Ccl19* (N = 3 group). Data are ΔCp values relative to *Gapdh*. Means and SEM are represented. SAMR1 and SAMP8 vehicle groups were compared by Student t-test. Within SAMP8 cohorts, different drug treatments were compared by oneway-ANOVA with Dunnett and SNK post-hoc analysis. *p < 0.05, **p < 0.01, ***p < 0.001.

immediate-early genes (IEGs) like *Egr1/2*, *cFos*, *Npas4*, *Dusp1* and *Arc* [38–42] (Fig 5B; NCBI GEO: GSE100413). The role of IEGs is explored in more detail in the section on behavior below. The identification of biomarkers with a potential for translation to the clinical setting is of special interest to drug development programs. *S100a9*, *Baiap3*, *Npw* and *Prph* (Fig 5C–5F) were selected and validated by qRT-PCR on a subset of individual samples from Exp1; *Ccl19* was included as a genotype control (Fig 5G).

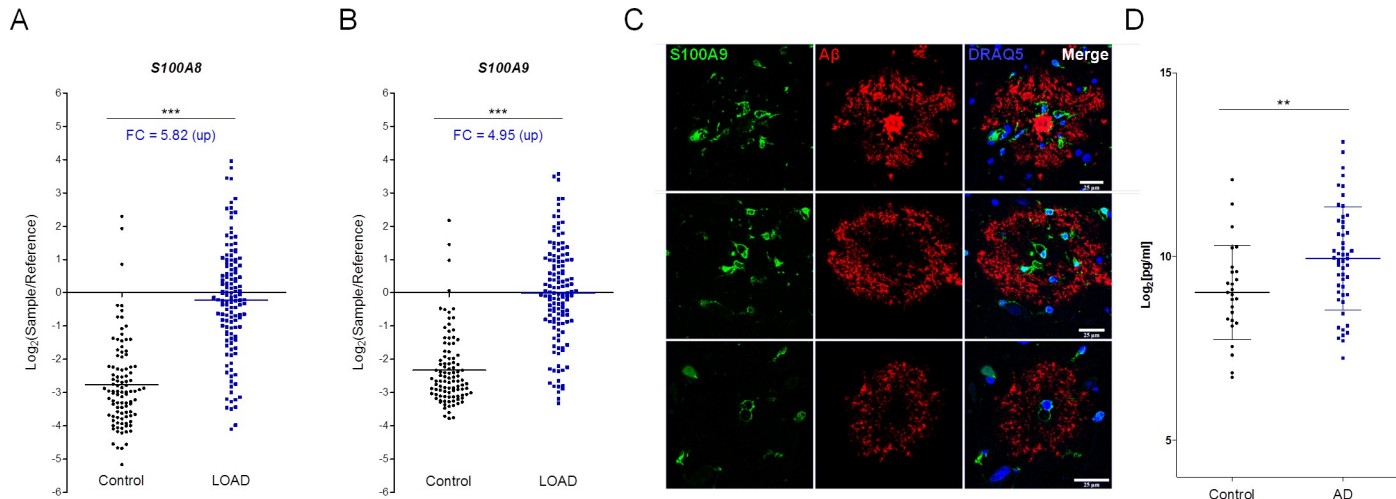

**Fig 6. S100A8 and S100A9 are differentially expressed in LOAD.** Re-examination of the expression of the orthologues of SAMP8 biomarkers in human prefrontal cortex in Control and LOAD samples from NCBI GEO GSE44770 [43]: (A) S100A8, (B) S100A9. Control: N = 101 and LOAD: N = 129 subjects. Samples are represented as $Log_2$ (sample/reference sample). Mean ± SD are represented. Fold changes (FC) were calculated as 2^[average $Log_2$(LOAD/Reference values)– average $Log_2$(Control/Reference values)]. Normality was verified by D'Agostino & Pearson omnibus normality test. If data passed the normality test statistical significance was calculated by unpaired t-test, otherwise the Mann-Whitney test was applied. $^*p < 0.05$, $^{**}p < 0.01$, $^{***}p < 0.001$. (C) Double-labeling immunofluorescence and confocal microscopy of paraffin sections with S100A9 (green, left), beta-amyloid (red, middle) and merged with DRAQ5^TM stained nuclei (blue, right) in temporal cortex of patients with Alzheimer's disease. Examples of S100A9-immunoreactive microglia localized in the vicinity of β-amyloid plaques. Bar = 25 μm. (D) S100A8/9 levels in CSF samples from AD patients (N = 51) and healthy controls (N = 26), as determined by S100A8/A9 ELISA. Data are expressed as $Log_2$ of the S100A8/A9 heterodimer concentration in pg/ml. Each dot represents one CSF donor and mean ± SD of each group are shown. Fold Change was calculated as the ratio of the average S100A8/A9 concentrations of the AD and control sample groups: average ([S100A8/A9 pg/ml]$_{AD\ SAMPLES}$)/average ([S100A8/A9 pg/ml]$_{CONTROL\ SAMPLES}$). Normality was verified by D'Agostino & Pearson omnibus normality test and statistical significance was calculated by unpaired t-test ($t(75) = 2.810$; $p = 0.0063$). $^{**}p < 0.01$.

Analysis of previously published GE data in Late Onset Alzheimer's Disease (LOAD) (NCBI GEO: GSE44770, [43]), show that *S100A8* and *S100A9* are among the highest up-regulated genes in LOAD (Fig 6A and 6B). Furthermore, S100A9 and the S100A8/9 heterodimer had been described to be augmented respectively in the brain and CSF of AD patients [44], although the sample number was quite small. To confirm these findings, we analyzed S100A9 protein expression in human brain sections. Staining was heterogeneous but frequently located in microglia residing in close proximity, surrounding or surrounded by abeta in AD samples (Fig 6C). Using a collection of biobank CSF samples, we confirmed the S100A8/9 heterodimer is significantly up-regulated in AD patients relative to controls (Fig 6D) and therefore an excellent candidate to monitor the pharmacological activity of ORY-2001 in CSF in clinical trials.

## ORY-2001 does not affect anxiety

SAMP8 mice have been reported to have age dependent modified anxiety [45]. To evaluate whether ORY-2001 affected anxiety, male mice were treated from month 5 of age with vehicle, 0.32 or 0.96 mg/kg/day ORY-2001, or 3 mg/kg/day RSG. In month 7 the animals were sequentially submitted to the open field (OF) and elevated plus maze (EPM) test to evaluate the effect on anxiety; and in the NORT. NORT results were essentially as in previous experiments. There were no significant genotype or treatment differences in the OF test (Fig 7A). SAMP8 mice spent a significantly increased time in the open arms of the EPM relative to SAMR1 mice, but this behavior was not significantly modified by ORY-2001 (Fig 7B). Therefore, ORY-2001 did not have anxiolytic or sedative activity in SAMP8 mice.

ORY-2001 was also evaluated in the rat rearing isolation model. Adult isolated male rats at PND 61 were treated with vehicle or 0.16 or 0.48 mg/kg/day ORY-2001 administered in

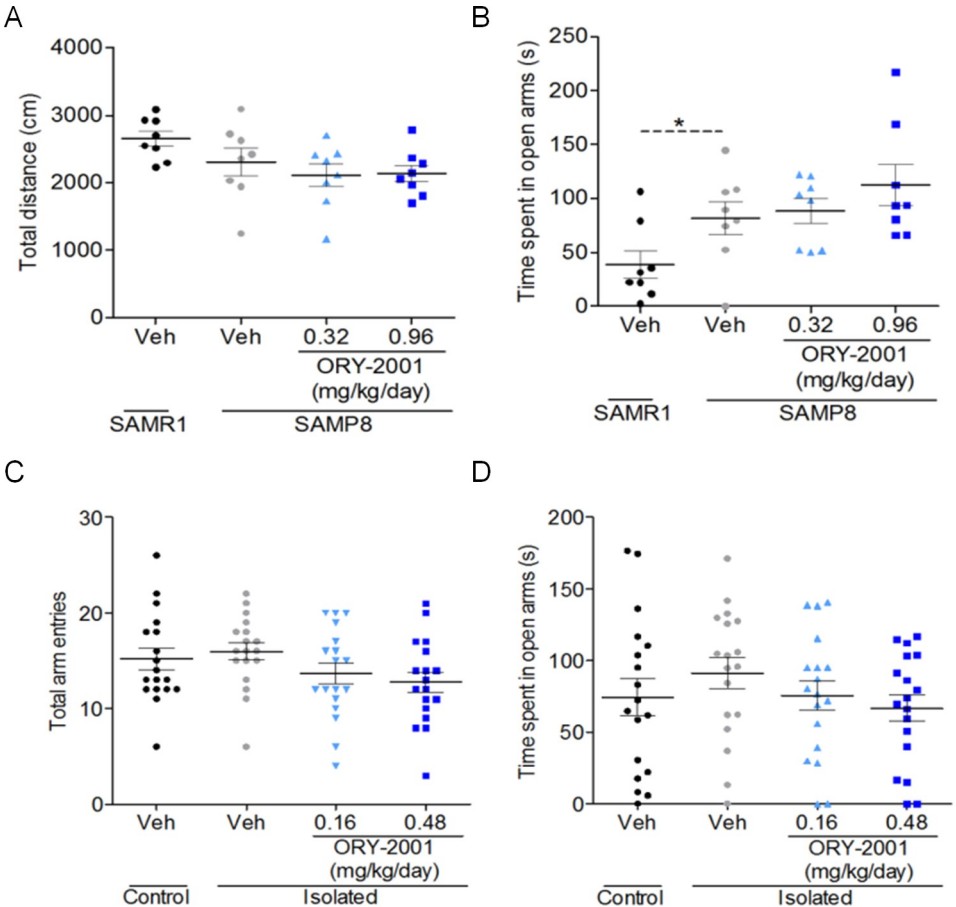

**Fig 7. ORY-2001 does not have anxiolytic or sedative effects.** Effect of ORY-2001 on SAMP8 mice: (A) Locomotor activity as assessed by total distance travelled in the open field test (N = 8/group), (B) anxiety as assessed by the time spent in the open arms in the elevated plus maze (t(14) = 2.183; p = 0.0465; N = 8/group), Effect of ORY-2001 on rats in the isolation rearing model: (C) Locomotor activity (N = 18/group) and (D) anxiety in the Elevated Plus Maze (N = 18/group); Means and SEM are represented. SAMR1 and SAMP8 vehicle or Non Isolated and Isolated groups were compared by Student's t-test. Within SAMP8 mice or Isolated groups, different drug treatments were compared by t-test or one way-ANOVA with Dunnett and SNK post-hoc analysis. *p < 0.05, **p < 0.01, ***p < 0.001.

drinking water for approximately 1 month. Effect on anxiety was evaluated in the EPM (PND 87–88) and on aggressiveness and social interaction in the RI test (PND 94). ORY-2001 did not produce any significant effect on anxiety or locomotor activity as assessed in the EPM (Fig 7C and 7D).

## ORY-2001 reduces aggression and improves social behavior

Impaired sociability and other social behavior alterations are associated with many psychiatric and neurodegenerative disorders.

SAMP8 mice exhibit aggressive behavior, which to the best of our knowledge had not been evaluated previously. The effect of ORY-2001 on aggression and social interaction was evaluated in the resident-intruder (RI) test (Exp8). A first cohort of SAMP8 animals was treated with vehicle, 0.32 or 0.96 mg/kg/day ORY-2001 for 5 weeks from month 5 of age; tested in the NORT and in the RI test, in which an intruder mouse of different strain is introduced in the cage of the tested animal (the "stress" cohort). A second cohort (the "basal" cohort) was treated pharmacologically in identical manner but not subjected to the RI test, and included as a

comparator for subsequent biomarker analysis (see below). SAMR1 mice were included as a reference strain in both cohorts. No significant strain- or treatment-differences were found in the time spent in social interactions or in the number of rearings in male mice (S4A Fig). SAMP8 mice showed a tendency to increase the latency to attack after ORY-2001 treatment (S4B Fig). The number of attacks and especially number of clinch attacks increased strikingly in SAMP8 relative to SAMR1 mice, and was lowered to SAMR1 levels by ORY-2001, in absence of any sign of sedation (Fig 8A, S4C Fig, S1 Video). This experiment was repeated using ORY-2001 at 0.96 mg/kg/day with similar results (Exp5, S3 Table). These data showed ORY-2001 reduced aggressiveness of SAMP8 mice, without causing sedation.

SAMP8 mice have been shown to have deficits in social interaction [45]. We evaluated the effect of 0.96 mg/kg/day ORY-2001, initiated at 8 months of age, on social interaction alter-ations of 12 month old female aged mice in the Three-Chamber Test (TCT, Exp7). This test is useful for quantifying effects in social interaction in animals exhibiting innate or acquired defi-cits in social behavior. Contrary to female SAMR1 mice, SAMP8 mice showed no preference for the socialization chamber in which a mouse is introduced over a chamber with an unani-mated object, and they also spent less time exploring the cage with a mouse. Treatment restored the time SAMP8 spent exploring mice introduced in the cage (Fig 8B), and the prefer-ence for the socialization chamber (S4D Fig and S2 Video) to SAMR1 levels.

We also studied the effect of ORY-2001 in the rat isolation rearing model. Rats were isolated after weaning on Post Natal Day (PND) 21 and deprived of the normal environment that pre-conditions their social behavior. Isolation in this phase of the development of the rat leads to pathological aggressive behavior [46] and/or a lack of interest for social interactions [47]. The rat isolation model has been proposed to model social avoidance occurring in neuropsychiatric disorders [48]. Adult isolated male rats at PND 61 were treated with vehicle or 0.16 or 0.48 mg/kg/day ORY-2001 administered in drinking water for approximately 1 month and evalu-ated in the RI test at PND 94. The RI test did not reveal aggressive behavior in the rats nor did isolation significantly affect active or passive social interaction (S4E and S4F Fig). The most striking behavioral difference between non-isolated and isolated rats in the RI test was seen in the social avoidance parameters. The time without social interaction in isolated rats was dose dependently reduced by treatment (Fig 8C), and the number of evitations, which was greatly increased in isolated rats, was restored to normality by treatment with ORY-2001 (Fig 8D).

Summarized, ORY-2001 drastically reduced aggressiveness and improved social interaction alterations in these rodent models, but did not act as a sedative or anxiolytic drug.

**ORY-2001 modulates the GE response in the PFC.** The prefrontal cortex (PFC) is known to play an important role in the control of aggressive behavior. We performed a genome-wide microarray-based survey on pooled PFC samples from SAMR1 mice and vehicle and ORY-2001 treated SAMP8 mice from the "basal" cohort. All samples were compared to the sample of the vehicle treated SAMP8 mice, using two-color hybridization. Revision of the genotype specific expression signature confirmed the Chr 4 genes *Ccl19*, *Ccl27*, *LOC664574* were expressed lower; and *Kif5b* and *Pppr1a1* were expressed higher in the SAMR1 relative to SAMP8 pools (Fig 9A). The survey further revealed that treatment of SAMP8 mice up-regu-lated *Rbm3* and *Rwdd2*. The genes *Baiap3*, *Npw*, *Rspo1*, *Kremen* and *Doc2a*; previously identi-fied in the hippocampus, were mildly increased. *Pcdh21*, *Doc2g* and *Pbx3* were down-regulated by treatment in the basal situation in the PFC, as were the immediate-early genes *Fos*, *Npas4*, *Tac1*, *and Egr1/2*; GABAergic genes relevant to synaptic plasticity such as *Calb2 and Gad1*; genes involved in signal transduction such as *Gng4 and Doc2g* [49–51], or neuro-peptides like *Penk1*, involved in pain signaling and anhedonia [52] (Fig 9B). Many of the genes down-regulated by ORY-2001 were over-expressed in vehicle treated SAMP8 vs SAMR1 mice, therefore treatment at least partially rebalanced the expression profile.

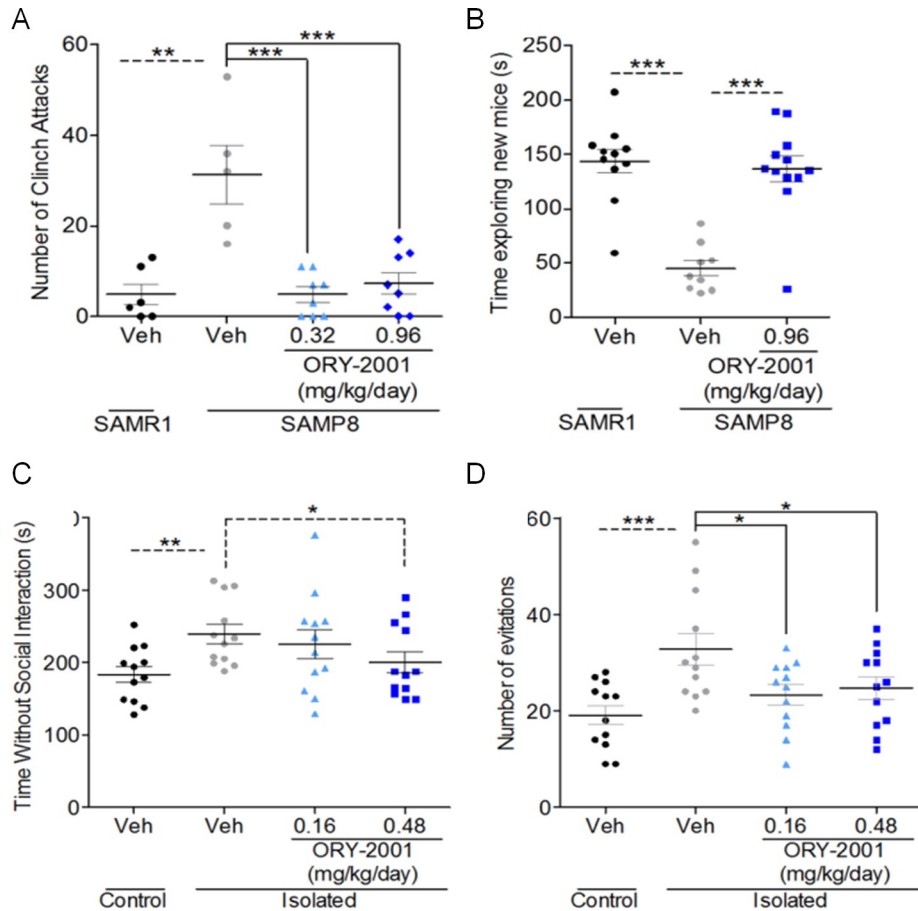

**Fig 8. Inhibition of KDM1A with ORY-2001 remedies behavior alterations.** Effect of ORY-2001 on SAMP8 mice:
(A) aggressive behavior as assessed by the total number of clinch attacks in the resident intruder test ($F_{2,18} = 16.76$;
$p < 0.0001$; N = 5–8 males/ group) and (B) social interest as assessed by the time exploring the new mice in the three
chamber test ($t(19) = 6.008$; $p < 0.0001$; N = 9–12 females/group). Effect of ORY-2001 on social behavior of rats in the
isolation rearing model in the Resident Intruder test reflected by (C) the time spent without social interaction and (D)
the number of active avoidances. ORY-2001 reduced both the time without social interaction ($t(22) = 2.094$; $p = 0.048$)
and the number of social avoidances ($F_{2,33} = 4.941$; $p = 0.013$) (N = 12/group). Means and SEM are represented.
SAMR1 and SAMP8 or Non Isolated and Isolated vehicle groups were compared by t-Test. Among the SAMP8 or
Isolated groups, different drug treatments were compared by one way-ANOVA with Dunnett and SNK post-hoc
analysis. *$p < 0.05$, **$p < 0.01$, ***$p < 0.001$.

The expression of selected genes was also analyzed in individual mice from the different
cohorts. The genotype and treatment group of each individual animal was verified using *Ccl19*
qRT-PCR (Fig 9C) and KDM1A TE analysis (Fig 9D), respectively.

It is known that the stress response encompasses modulation of plasticity related genes in a
neuronal activity dependent fashion. qRT-PCR analysis revealed that the stress response of
IEGs like *Fos* and *Npas4* and of other genes like *Calb2* and *Gng4* (Fig 9E–9H) is compromised
in SAMP8 relative to SAMR1 mice, and that treatment with ORY-2001 rescued or potentiated
the response capacity, by increasing the gap between the "basal" and "stress" induced levels in
the PFC of SAMP8 mice. This effect was reminiscent of the effect of the broad spectrum
HDAC inhibitor SAHA on the expression of *Fos* and *Egr1* in KDM1A-E8a KO mice. These
mice display a deficient stress response and a low anxiety phenotype that can be rescued by
SAHA [53].

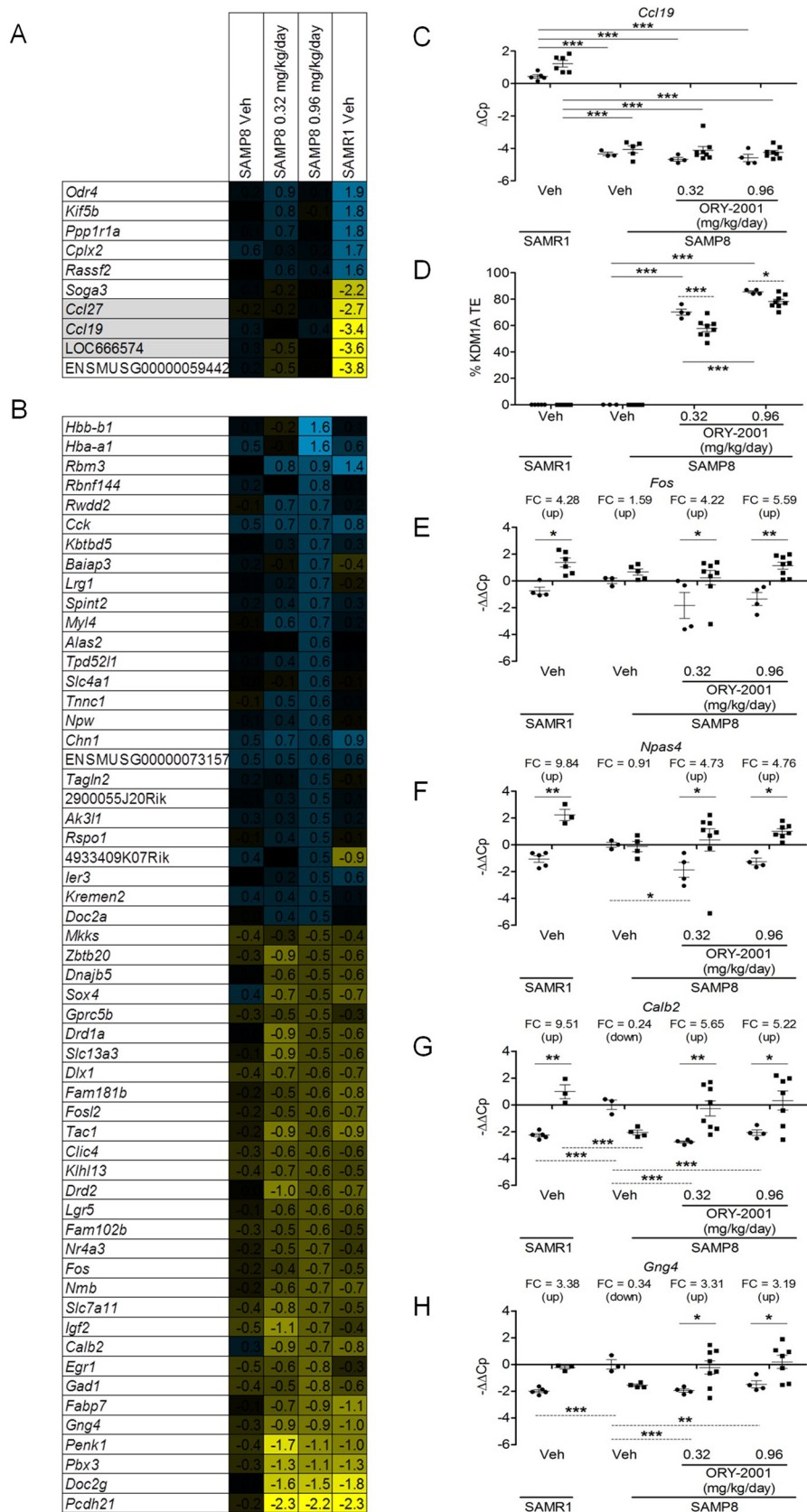

**Fig 9. ORY-2001 restores the IEG stress response in the SAMP8 prefrontal cortex.** Gene expression changes in SAMP8 mice: (A) Top 5 differentially regulated genes (up and down) in the PFC of SAMP8 vs SAMR1 mice in the basal condition identified by microarray survey. *Ccl19*, *Ccl27* on chr4 amplified in SAMP8 compared to SAMR1 mice are genotype controls and highlighted in grey. (B) Microarray survey performed on PFC tissue of vehicle- and ORY-2001-treated mice shows that ORY-2001 partially restores the GE pattern seen in SAMP8 mice to SAMR1 controls. GE changes in basal conditions following treatment with ORY-2001 (genes with an expression change $Log_2 > 0.5$ or $Log_2 < -0.5$ (1.4 fold up or down) at the dose of 0.96 mg/kg/day in relation to vehicle-treated SAMP8 mice. Genes are ranked according to the highest expression change in the ORY-2001 0.96 mg/kg/day condition. All values represent $Log_2$(Fold Change). The survey was performed on pooled samples from each experimental group (N = 3–5 mice per group). Veh = Vehicle. (C) qRT-PCR analysis of *Ccl19* expression performed on cortex tissue was used to verify the genotype of individual mice. Dots: basal; squares: stress cohorts. Data were normalized by *GusB* and are plotted as ΔCp values. Graphs represent mean ± SEM. Solid line: one-way ANOVA analysis of basal and stress levels using Dunnett's multiple comparison test relative to the vehicle-treated SAMR1 condition. ***p < 0.001.(D) KDM1A TE analysis used to verify the treatment group of individual mice. Dots: basal; squares: stress cohorts. Graphs represent mean ± SEM. Solid line: one-way ANOVA analysis of basal and stress levels using Tukey's multiple comparison test. Dotted line: two-way ANOVA with Bonferroni post-hoc analysis. *p < 0.05, ***p < 0.001. The effect of ORY-2001 treatment on the expression in the prefrontal cortex, following exposure to stress of (E) *Fos* ($F_{3,34}$ = 1.128; p = 0.3514 Interaction), (F) *Npas4* ($F_{3,30}$ = 2.312; p = 0.0961), (G) *Calb2* ($F_{3,30}$ = 7.361; p = 0.0008) and (H) *Gng4* ($F_{3,30}$ = 5.648; p = 0.0034). Basal: N = 3–5 per group; RI test: N = 3–8 per group). Dots: basal; squares: stress cohorts. All qRT-PCR data are plotted as -ΔΔCp values relative to the basal vehicle-treated SAMP8 group. Fold changes (FC) were calculated as 2^[average (stress -ΔΔCp values)—average(basal -ΔΔCp values)]. *Fos* data were normalized by *Gapdh*; *Npas4*, *Calb2* and *Gng4* by *GusB*. Graphs represent mean ± SEM. Solid line: two-way ANOVA with Bonferroni post-hoc analysis. Dotted line: one-way ANOVA analysis of basal or stress levels using Dunnett's multiple comparison test relative to the vehicle-treated SAMP8 condition. *p < 0.05, **p < 0.01, ***p < 0.001.

To evaluate the relevance of these markers to Late Onset Alzheimer's Disease (LOAD), we also re-examined the biomarkers in the NCBI GEO: GSE44770 dataset [44]. While it is evidently not possible to analyze the stress response in post-mortem samples, it was noted that IEGs, GABAergic or glutamatergic synapse genes and signal transduction genes) that exhibited deficient stress response in SAMP8, as well as neuronal plasticity genes like *UCHL1* and *RBM3*; and the neuropeptide *PENK1* (Fig 10, S5 Fig) were down-regulated in LOAD vs normal samples. Together, these data indicate that biomarkers modulated in SAMP8 mice may be of relevance to Alzheimer's disease.

## KDM1A is recruited by transcription factors that modulate IEGs

The effects of KDM1A inhibition on transcription in different cell types or tissues may be conditioned largely by the endogenous TF landscape. A dual strategy was used to start and unravel the KDM1A interaction network in human neuronal (progenitor) cells or tissues, including unbiased and candidate based approaches. KDM1A chemoproteomics [8,25] was used to pull down (S6 Fig) and subsequently analyze KDM1A complexes from undifferentiated SH-SY5Y cells by tandem mass spectrometry (S4 Table). This approach identified many components of the core complex including KDM1A, RCOR3 > RCOR2 > RCOR1, HDAC2 > HDAC1, and GSE1; in addition to KDM1A recruiting ZNF TFs like ZMYM2; ZNF516; and ZNF217 (Fig 11A). No SNAG domain TFs (GFI1, GFI1B, SNAI, INMS1, OVOL2) were identified using chemoproteomics, as expected since chemoprobe and SNAG domain binding is mutually exclusive [8] Chemoprobe- and KDM1A-interactor ELISAs confirmed the interaction with HDAC2 and HDAC1 (Fig 11B). Interactor ELISAs further detected the ZNF TFs REST, MYT1, ZMYM3, ZEB1, ZNF326 and the SNAG domain containing ZNF TFs OVOL2, SNAI1; the general transcription factor GTF2I, as well as SRF, CTBP1, SIRT1 and SVIL. The ELISA assays revealed striking differences between the KDM1A interactome in SH-SY5Y cells (REST, ZNF217, MYT1, ZMYM2, and CtBP1 high and SRF low) and human hippocampus (SRF, OVOL2, GTF2I, and ZNF217 high, REST, MYT1, ZMYM2 and CTBP1, SVIL low) (Fig 11C and 11D). SRF was previously described to interact with KDM1A in the mouse hippocampus and is implicated in the transcriptional control of IEGs including of *c-Fos* [53]. GTF2I is

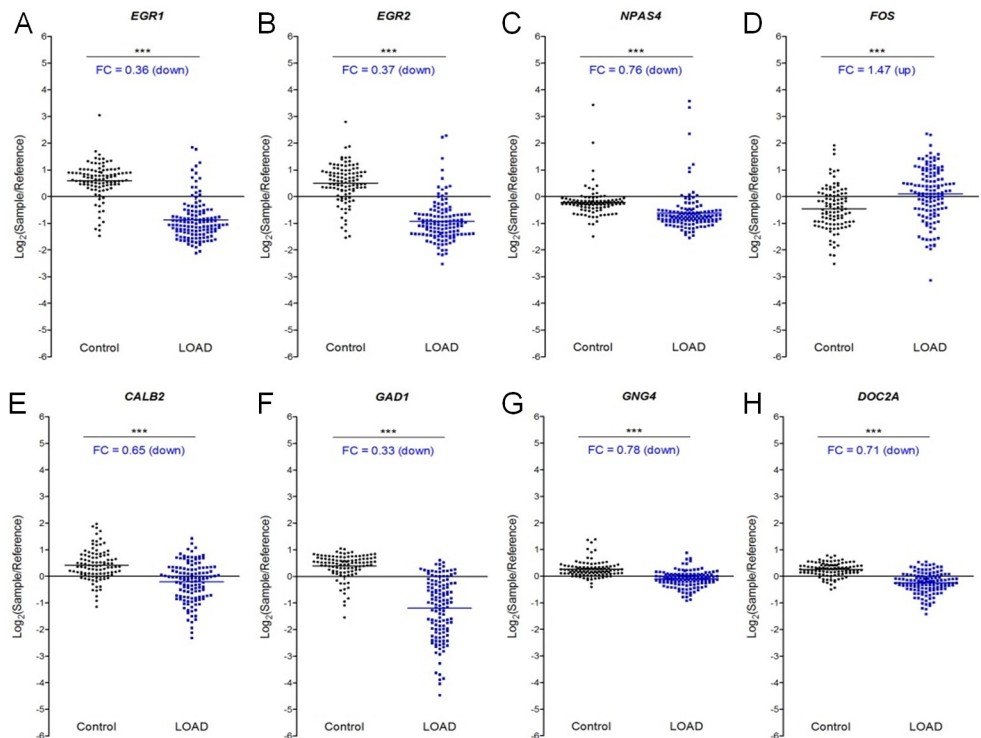

**Fig 10. Biomarkers modulated by ORY-2001 are differentially expressed in LOAD.** Re-examination of the expression of the orthologues of SAMP8 biomarkers in human prefrontal cortex in Control and LOAD samples from NCBI GEO GSE44770 [44] reveals differential expression of immediate early genes (A) *ERG1*, (B) *ERG2*, (C) *NPAS4* and (D) *FOS*, (E) *CALB2*, (F) *GAD1*, (G) *GNG4* and (H) *DOC2A*. Control: N = 101, average age 62.1 years, male to female ratio = 4.3; LOAD: N = 129 subjects, average age 80.1 years, male to female ratio = 0.9. All samples are represented as Log$_2$ (sample/reference sample) calculated from GSE44770 data. Mean ± SD are represented. Fold changes (FC) were calculated as 2^[average Log$_2$(LOAD/Reference values)–average Log$_2$(Control/Reference values)]. Normality was verified by D'Agostino & Pearson omnibus normality test. If data passed the normality test statistical significance was calculated by unpaired t-test, otherwise the Mann-Whitney test was applied. $^*$p < 0.05, $^{**}$p < 0.01, $^{***}$p < 0.001.

involved in behavior and neurodevelopment disorders [54], has been reported to interact with KDM1A in human iPSCs [55] and, in another context, to be involved in the control of IEGs [56]. The interactions of KDM1A with SRF and GTF2I may therefore explain the modulation of IEGs in the mouse brain by ORY-2001 treatment.

## Discussion

The use of epigenetic modulators offers a unique opportunity to redress transcriptional imbalances observed in neurodegenerative disease. HDAC inhibitors have been reported to be beneficial in animal models of a variety of human neurodegenerative conditions [57]. Unfortunately, few HDAC inhibitors have progressed in the clinic for use in indications beyond hemato-oncology; hampered by the pleiotropy of HDAC function, the broad cytotoxicity of most HDAC inhibitors, and the difficulty to produce isoform selective inhibitors of HDAC2. Attempts to design these were reported to be underway [58] yet recently efforts have been reoriented towards the development of RCOR1 complex-selective rather than isoform-selective Class I HDAC inhibitors [59].

KDM1A and Class I HDACs coincide in transcription regulation complexes, more precisely in the RCOR1 complex [60]. We chose KDM1A as the prime target and developed ORY-2001.

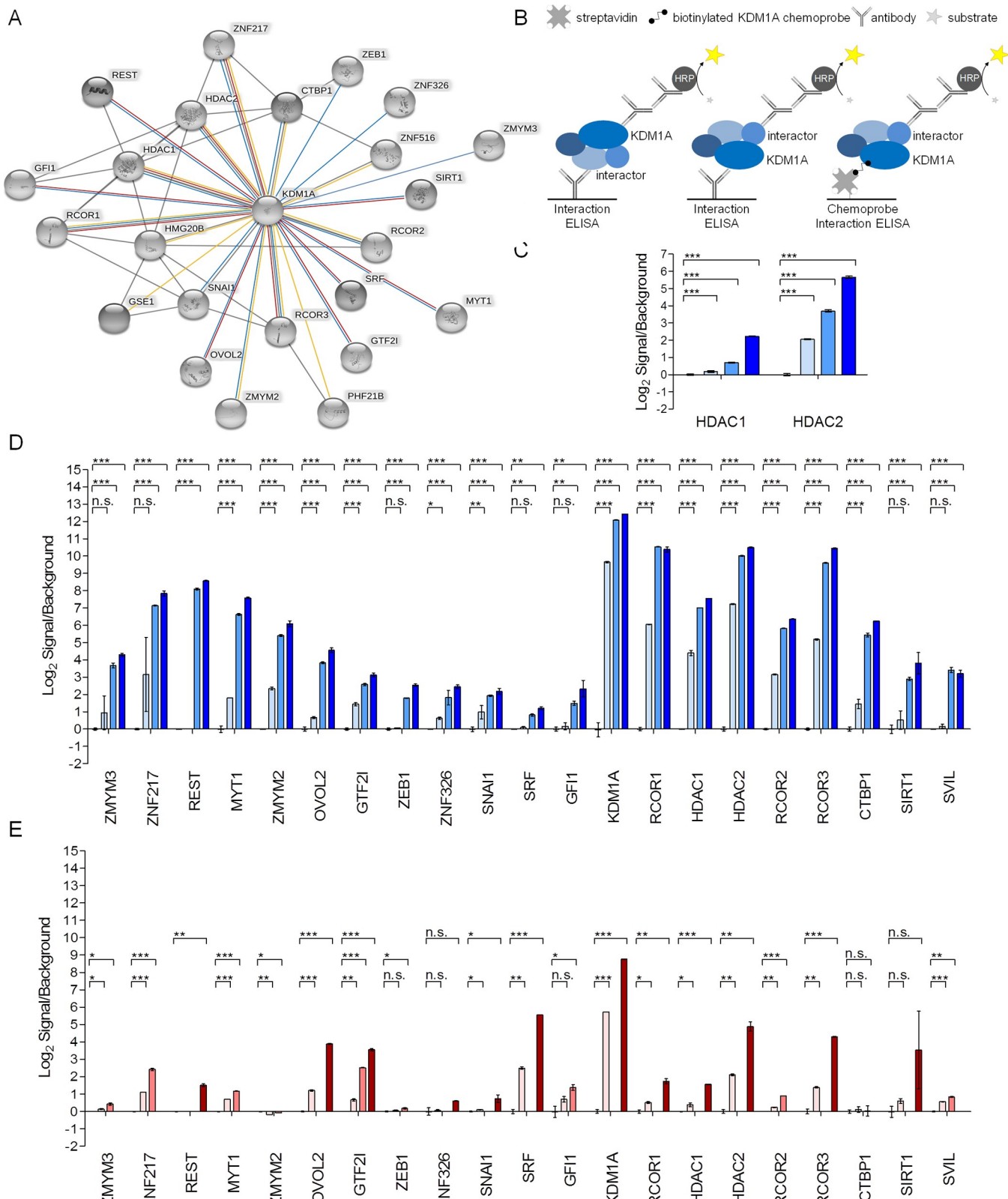

**Fig 11. KDM1A is recruited by transcription factors that modulate IEGs.** (A) Schematic representation of KDM1A-interacting proteins detected by mass spectrometry and by ELISA. The figure was generated using STRING. Blue line: interaction detected by ELISA on SH-SY5Y cells; red line: interaction detected

by ELISA on human hippocampal tissue; orange line: interaction detected by mass spectrometry; grey line: protein-protein interaction displayed by STRING. (B) Schematic representation of KDM1A interaction ELISAs (left) and chemoprobe interaction ELISA (right). The KDM1A-containing multiprotein complexes are captured either by an antibody directed against one of its components or by the OG-881 chemoprobe; and detected using an antibody directed against KDM1A or another component of the complex. (C) Chemoprobe- interactor ELISAs in SH-SY5Y cells. OG-881 chemoprobe-bound KDM1A complex was captured on streptavidin coated plates and detected using with an HDAC1 or HDAC2 antibody. (D) ELISA on SH-SY5Y cells (N = 1; n = 3 [n = 2 for ZMYM3, ZNF217, REST, SRF, GFI1, HDAC1, HDAC2]). Protein amounts used were 2.5, 25, or 44 µg per well (light, mid, dark blue). (E) ELISA on human post-mortem hippocampal tissue from healthy controls (N = 1; n = 2). Protein amounts were 10, 30, or 80 µg per well (light, mid, dark red). Data in b-d represent Log2 values of the relative luminescence signal (RLU) normalized to the corresponding background. Mean ± SD is shown. The statistical significance was calculated by oneway-ANOVA with Dunnett multiple comparison test relative to the negative control condition (when n = 3) or by Student's t-test relative to the negative control condition (when n = 2). $^*$p < 0.05, $^{**}$p < 0.01, $^{***}$p < 0.001.

KDM1A inhibitors have been described to disrupt the interaction of KDM1A and GFI1 [61], a SNAG domain containing ZNF transcription factor key involved in hematopoiesis [62]. The impact of long term treatment with a KDM1A inhibitor on hematology was a potential concern. MTD studies showed that at sufficiently high doses, ORY-2001 indeed impacts hematology. In spite of its effects on hematology at high doses, efficacy studies in rodent models showed that it provides a good therapeutic window for treatment of CNS diseases.

ORY-2001 can rescue the profound memory impairment developed by SAMP8 mice at doses that have no impact on hematology. Positive effects on memory have been reported with other KDM1A inhibitors. T-448 improved the learning function in NMDA receptor NR1 subunit hypofunction mice [63]. Iadademstat (ORY-1001), a highly potent selective KDM1A inhibitor in development for treatment of oncological diseases [8], reversed cortical synaptic plasticity deficit and the working memory of SETD1 +/- mice, a model for schizophrenia [64]. These data provide support for the potential of KDM1A inhibitors in learning and memory.

On the other side, a single high dose of the KDM1A inhibitor RN-1 was reported to provoke deleterious effects on long term memory [65]. However, at the very high dose used in that study, RN-1, a compound originally disclosed by Oryzon [66], is also a non-selective muscarinic receptor antagonist (MRA) (S4 Table). Since MRAs like scopolamine are known to induce amnesia in animals and humans [67], this activity may provide an alternative explanation to the observed effects. No detrimental neurological effects were reported when using lower doses of RN-1 in long term studies in baboons [68].

ORY-2001 is not an MRA (S1 Table) and did not lead to neurological alterations in the Functional Observation Battery after administration of a single dose to rats of up to 100 mg/ kg, 500 times the NoAEL for long term studies.

Our data show the *in vivo* pharmacological effect of ORY-2001 can be attributed primarily to KDM1A modulation. The genome wide GE analysis showed that ORY-2001 reduced an inflammatory signature and modulated the expression of genes, including immediate early genes (IEGs), previously described to be required for memory, synaptic plasticity and cognition [69–71]. Interference with the organism´s capacity to induce these genes provokes memory deficit, as can be observed in mice with attenuated levels of *Acetyl-CoA synthetase 2* (*Acss2*), a gene that regulates histone acetylation [72] or interestingly also in the selective knockout of the neurospecific KDM1A-E8a splice form [5]. The original hypothesis on KDM1A-E8a postulated that *in vivo*, this splice form acts as a dominant negative form that counteracts the repressive activity of the canonical KDM1A splice form [14, 17, and 73]. Alternatively it was proposed that the neurospecific splice form has a distinct substrate specificity [8, 18].

Inhibition of KDM1A by ORY-2001 appears to have the opposite effect of the deletion of KDM1A-E8a, a splice form required for proper memory function. Indeed, contrary to what happens in KDM1A-E8a KO mice, where the IEG response is compromised; we have shown that chronic pharmacological modulation of KDM1A by ORY-2001 in SAMP8 mice *facilitates* the response of IEGs and shifts the epigenetic balance in a manner favorable to memory. SRF

and GTF2I, two factors known to be involved in the control of transcription of IEGs including *cFos* [74, 56], form part of the brain KDM1A interactome and are involved in memory and/or behavior [75, 54].

ORY-2001 positively affects cognition but it also corrected behavioral changes, like aggressive behavior and social interaction deficits. These data illustrate the potential of ORY-2001 for the non-sedative treatment of behavior disturbances associated with a variety of neuropsychiatric disorders in addition to neurodegenerative diseases.

ORY-2001 also reduced the expression of an inflammation signature in SAMP8 mice. The amplifier of inflammation *S100a9* appeared especially interesting as a biomarker. Knockout or knockdown of *S100a9* decreases the development of memory impairment and neuropathology in transgenic models of AD mice; in humans S100A9 is up-regulated in the hippocampus of AD patients and expressed in microglia in the proximity to Aβ plaques [76, 45], and has been suggested to act as a seeding factor for Abeta accumulation [77]. Strikingly, *S100A8* and *S100A9* are among the top up-regulated genes in the PFC of LOAD patients; and the S100A8/9 heterodimer is increased in the CSF of AD patients. *S100a8* and *S100a9* were reported to be highly expressed in FACS purified CD11B+CD11C- microglia, associated with the inflammatory response in a mouse model of AD, as opposed to CD11B+CD11C+ microglia, postulated to have a protective immune-modulatory phenotype [78]. Similarly, disease associated microglia (DAM) cells, assumed to have a protective function, have low *S100a8* and *S100a9* levels; while granulocytes/neutrophils have high *S100a8* and *S100a9* expression [79] and may promote AD-like pathology and cognitive decline [80].

S100A9 is also up-regulated in other diseases with a neuro-inflammatory component, including traumatic brain injury (TBI) [81], post-operative cognitive dysfunction (POCD) [82], and multiple sclerosis (MS) [83]; broadening the potential therapeutic applications of ORY-2001. Indeed, we recently tested ORY-2001 in preclinical models for MS and found it to exert potent therapeutic activity, although the effect is unlikely to be attributed exclusively to S100A9.

In AD patients, cognitive decline is often accompanied by aggressiveness, agitation, psychosis, depression and apathy. Our data demonstrate that ORY-2001 acts on two processes that are frequently affected in neurodegenerative diseases: cognition and behavior. These findings position the compound as a promising candidate for the treatment of AD. In view of recent evidence for potential implication of herpes type viruses in the development of AD [84], it is worthwhile to mention that KDM1A inhibitors can block herpes virus lytic replication, reactivation from latency in vitro, as well as infection, shedding and recurrence in animal models [85–87], although ORY-2001 has not been evaluated in this context.

## Conclusions

Vafidemstat (ORY-2001) is a brain penetrant, orally bioavailable, small molecule inhibitor of the histone lysine demethylase KDM1A. ORY-2001 corrects memory deficit and behavior alterations including aggression and social interaction deficits in SAMP8 mice and social avoidance in the rat rearing isolation model. ORY-2001 increases the responsiveness of IEGs, induces genes required for cognitive function and reduces neuroinflammation in the brain of mouse models of CNS disease. Interestingly, many key genes modulated by ORY-2001 are differentially expressed in the brain of patients with AD or other diseases of the central nervous system.

The preclinical data presented in this manuscript provide support for vafidemstat (ORY-2001) for treatment of memory deficit and behavior alterations in neurodegenerative and psychiatric diseases.

Safety, tolerability, pharmacokinetics, pharmacodynamics and brain penetration of vafidemstat have been evaluated in a Phase I trial in healthy young and elderly volunteers

(EUDRACT Nº 2015-003721-33). Currently vafidemstat is advancing in Phase IIa trials in patients with relapse-remitting and secondary progressive MS patients (SATEEN; EUDRACT Nº 2017-002838-23), with mild to moderate Alzheimer's disease (ETHERAL; EUDRACT Nº 2017-004893-32 and ETHERAL-US; NCT03867253), and in patients with aggression across several psychiatric indications and in AD (REIMAGINE and REIMAGINE-AD; EUDRACT Nºs 2018-002140-88 and 2019-001436-54). The first positive human efficacy data for treatment of aggression in Attention Deficit and Hyperactivity Disorder, Autism Spectrum Disorder (ASD) and Borderline Personality Disorder have been reported, providing support for vafidemstat as an emerging therapeutic option [88]. Finally, recent data suggest that specific genetically defined populations of patients with schizophrenia and ASD may also benefit from treatment with vafidemstat.

## Supporting information

**S1 Fig. Primary pharmacological properties of ORY-2001.** (A) Bubble graph representing KDM1A, MAO-B and MAO-A $IC_{50}$ values inhibitors. MAO-B and KDM1A are represented on the x and y-axis, and the MAO-A activity is represented as the size of the bubble. The dual KDM1A/MAO-B inhibitor ORY-2001 (blue), selective KDM1A inhibitor ORY-LSD1 (red), MAO-B inhibitors rasagiline(RSG) and selegiline (SLG) (green) and dual MAO-A/B inhibitor tranylcypromine (TCP, orange) have been labeled in the graph. (B) Time course of the change in absorption at 450 nM due to binding of ORY-2001 (40 μM) to the FAD co-factor in KDM1A (15 μM). (C) Maldi-Tof analysis confirms adduct formation between ORY-2001 and KDM1A.
(TIF)

**S2 Fig. ORY-2001 inhibits KDM1A > MAO-B > MAO-A *in vivo*.** *KDM1A inhibition*: (A) Dose and time response of platelet levels in male Wistar rats during and after 5 days of treatment with ORY-2001 at ●0.06 mg/kg (N = 3), ■0.2 mg/kg (N = 6), ▲0.4 mg/kg (N = 3), ▼0.6 mg/kg (N = 6), ◆2 mg/kg (N = 6), o 6 mg/kg (N = 6), □ 20 mg/kg (N = 6). (B) KDM1A Target Engagement analysis in rat. Top: schematic representation of the sandwich ELISA assay used for the determination of total and free KDM1A. Bottom: dose- and time-dependent response of KDM1A Target Engagement (%) measured in brain (solid line) and in PBMCs (dotted line) in male Sprague-Dawley rats treated for 5 days with ORY-2001 at ● (light blue) 0.06 mg/kg (N = 3) or ▲ (dark blue) 0.4 mg/kg (N = 3). *MAO-B inhibition*: Preventive effect of i.p. administration of ORY-2001 in an MPTP induced Parkinsonism model in C57BL/6 mice on (C) body weight: ORY-2001 (blue) at ▲ 0.3 mg/kg ▼ 1 mg/kg ◆3.0 mg/kg; RSG (green) at ■ 3 mg/kg, with ○ Control without MPTP, or ● Control vehicle + MPTP; (D) locomotor effect- horizontal distance; (E) TH+ neurons (N = 8/group); (F) representative image of histopathology. (G) Potentiation of PEA-induced symptoms in CD1 mice treated with ● ORY-2001 (blue), $ED_{50}$ = 6.4 mg/kg, ■ TCP (orange), $ED_{50}$ = 1.1 mg/kg, ▲ SLG (green), $ED_{50}$ = 8.0 mg/kg; cumulative score from N = 10/group. (H) *Ex vivo* measurement of MAO-B target inhibition in CD1 mice. Gray dot (acute), Black dot (chronic 5 days). *MAO-A inhibition*: (I) Tyramine pressor response in Wistar rats. Gray dot (acute), black dot (chronic 5 days); N = 6-7/group. (J) Potentiation of L-5-HTP-induced symptoms in C57BL/6 mice after treatment with ● ORY-2001 (blue), ■ TCP (orange), ▲ SLG (green); cumulative score from N = 10/group; (K) *Ex vivo* measurement MAO-A target inhibition in the brain of CD1 mice. Gray dot (acute), black dot (chronic 5 days). All treatments were QD per oral gavage unless stated otherwise.
Means ± SEM are represented. Different drug treatments were compared by Oneway-ANOVA with Dunnett analysis. $^{*}p < 0.05$, $^{**}p < 0.01$, $^{***}p < 0.001$.
(TIF)

**S3 Fig. ORY-2001 improves cognition in SAMP8 mice.** ORY-2001 treatment effect on the DI in the NORT in male SAMP8 mice. The retention test was evaluated 2 (A,B) and 24 (C,D) hours after training to measure changes in medium and long term memory. Five month old animals were divided in two groups receiving 2 (N = 15-16/group) (A,C) and 4 (N = 9-10/group) (B,D) months of treatment. Means and SEM are represented. SAMR1 and SAMP8 vehicle groups were compared by t-Test. Among the SAMP8 cohorts, different drug treatments were compared by oneway-ANOVA with Dunnett and SNK post-Hoc analysis. $^{**}p < 0.01$, $^{***}p < 0.001$.
(TIF)

**S4 Fig. ORY-2001 remedies social behavior alterations in SAMP8 mice and rats in the isolation rearing model.** Social behavior in SAMP8 mice: (A) number of rearings in the RI test performed, (B) latency to attack and (C) number of attacks of vehicle treated SAMR1 and vehicle or ORY-2001 treated SAMP8 male mice (N = 5-8/group). SAMP8 animals did not show significant differences in the number of rearings or latency to attack compared to SAMR1 mice but SAMP8 animals did show higher number of attacks. Treatment with ORY-2001 had no significant effect on the number of rearings in SAMP8, but a dose dependent tendency to increase in the latency to attack was observed and a clear effect to reduce the number of attacks. (D) In the Three Chamber Test, SAMP8 animals do not show preference for the chamber with the novel mice, treatment with ORY-2001 restored the normal preference to similar levels observed in the SAMR1 (N = 9-12/group). SAMR1 and SAMP8 were compared by t-TEST. Means and SEM are represented. Vehicle and drug vs vehicle treatments in SAMP8 mice or isolated rats were compared by oneway-ANOVA with Dunnett and SNK post-Hoc analysis. Means and SEM are represented. $^{*}p < 0.05$, $^{**}p < 0.01$, $^{***}p < 0.001$. Social behavior in the rat isolation rearing model: Time spent on (E) active and (F) passive social interactions in the RI test performed on vehicle treated control and vehicle or ORY-2001 treated isolated rats (N = 12/group). Isolated rats did not show significant differences in active or passive social interaction compared to non isolated rats and treatment with ORY-2001 had no effect on these parameters. Control and Isolated vehicle groups were compared by t-Test. Means and SEM are represented. Vehicle and drug vs vehicle treatments in SAMP8 mice or isolated rats were compared by oneway-ANOVA with Dunnett and SNK post-Hoc analysis. $^{*}p < 0.05$, $^{**}p < 0.01$, $^{***}p < 0.001$.
(TIF)

**S5 Fig. Biomarkers modulated by ORY-2001 are altered in Alzheimer's disease.** Re-examination of the expression of the orthologues of SAMP8 biomarkers shows differential expression of synaptic plasticity genes (E) *UCHL1* (F) *RBM3*, (G) *PENK* in human prefrontal cortex of Control and LOAD samples from NCBI GEO GSE44770 [44]. All samples are represented as Log$_2$ (sample/reference sample). Control: N = 101 and LOAD: N = 129 subjects. Means ± SD are represented. Fold changes (FC) were calculated as 2^[average Log$_2$ (LOAD/reference values)–average Log$_2$ (Control/reference values)]. Significance was calculated by Mann-Whitney test. $^{***}p < 0.001$.
(TIF)

**S6 Fig. Chemoprobe pulldown identifies components of the KDM1A complex in SH-SY5Y cells.** (A) Western blot of recombinant KDM1A and vehicle or ORY-2001 treated SH-SY5Y input and KDM1A chemoprobe pulldown (PD) samples, analyzed with anti-KDM1A (top) and anti-RCOR1 (bottom) antibodies. (B) Silver nitrate staining of recombinant KDM1A and vehicle or ORY-2001 treated SH-SY5Y input and KDM1A chemoprobe pulldown (PD) samples analysed by PAGE. C-: negative control (pulldown of vehicle treated cells in absence of

chemoprobe). 10% of the total pulldown was loaded per lane.
(TIF)

**S1 Table. ORY-2001 Pharmacological activity.** (A) Inhibition of FAD enzymes. (B) Inhibition of epigenetic enzymes. (C) Diversity screen.
(XLSX)

**S2 Table. Pharmacokinetics.** (A) ORY-2001. (B) ORY-LSD1.
(XLSX)

**S3 Table. Overview of SAM experiments.**
(XLSX)

**S4 Table. Chemoproteomics dataset.**
(XLSX)

**S5 Table. RN-1 (OG-98) Muscarinic receptor Off-Target Activity.**
(XLSX)

**S1 Video. ORY-2001 reduces aggression of SAMP8 mice.** Examples of the behavior of a vehicle treated male SAMR1 mouse (left), a vehicle treated male SAMP8 mouse (center) and a male SAMP8 mouse treated with 0.32 mg/kg/day of ORY-2001 (right) in the Resident Intruder test. SAMR1 and SAMP8 mice are white and the resident intruder is black. The videos fragments shown are cuts from the original recordings starting around min 7 of the test and played at 2X speed and for a total duration of 39 seconds (related to Fig 4C and S4C Fig).
(MP4)

**S2 Video. ORY-2001 promotes social behavior in SAMP8 mice.** Examples of the behavior of a vehicle treated female SAMP8 mouse (left) and a female SAMP8 mouse treated with 0.96 mg/kg/day of ORY-2001 (right) in the Three Chamber Test. The videos fragments shown are cuts from the original recordings starting and played at 8X speed and for a total duration of 45 seconds (related to Fig 4D).
(MP4)

**S1 File. Supporting data.**
(DOCX)

**S2 File. Supporting materials and methods.**
(DOCX)

## Acknowledgments

We are grateful to Manuela Cervelli and Paolo Mariottini from the University of Rome Tre for providing recombinant SMOX and to Andrea Mattevi and Claudia Binda from the University of Pavia for providing recombinant KDM1B and KDM1A splice forms.

## Author Contributions

**Conceptualization:** Tamara Maes, Matthew Fyfe, Julio César Castro Palomino, Carlos Buesa Arjol.

**Formal analysis:** Tamara Maes, Cristina Mascaró, David Rotllant, Michele Matteo Pio Lufino, Fernando Cavalcanti, Christian Griñan-Ferré, Mercè Pallàs, Roser Nadal, Antonio Armario, Isidro Ferrer, Alberto Ortega, Nuria Valls.

**Funding acquisition:** Tamara Maes, Carlos Buesa Arjol.

**Investigation:** Cristina Mascaró, David Rotllant, Michele Matteo Pio Lufino, Angels Estiarte, Nathalie Guibourt, Fernando Cavalcanti, Christian Griñan-Ferré, Mercè Pallàs, Roser Nadal, Antonio Armario, Isidro Ferrer, Alberto Ortega, Nuria Valls.

**Methodology:** Cristina Mascaró, Michele Matteo Pio Lufino, Nathalie Guibourt, Isidro Ferrer.

**Project administration:** Tamara Maes, Matthew Fyfe.

**Resources:** Christian Griñan-Ferré, Mercè Pallàs, Roser Nadal, Antonio Armario, Isidro Ferrer.

**Supervision:** Cristina Mascaró, David Rotllant, Angels Estiarte, Fernando Cavalcanti, Mercè Pallàs, Marc Martinell.

**Validation:** Cristina Mascaró, Michele Matteo Pio Lufino.

**Visualization:** Tamara Maes, David Rotllant, Alberto Ortega.

**Writing – original draft:** Tamara Maes, Cristina Mascaró, David Rotllant, Michele Matteo Pio Lufino, Fernando Cavalcanti.

**Writing – review & editing:** Tamara Maes, Cristina Mascaró, David Rotllant, Angels Estiarte, Nathalie Guibourt, Fernando Cavalcanti, Christian Griñan-Ferré, Mercè Pallàs, Roser Nadal, Antonio Armario, Isidro Ferrer, Alberto Ortega, Nuria Valls, Marc Martinell, Julio César Castro Palomino.

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
