## [Decision Letter · Decision Letter 0]

5 Feb 2020

PONE-D-20-00734

Modulation of KDM1A with vafidemstat rescues memory deficit and behavioral alterations.

PLOS ONE

Dear Dr. Maes,

Thank you for submitting your manuscript to PLOS ONE. After careful consideration, we feel that it has merit but does not fully meet PLOS ONE’s publication criteria as it currently stands. Therefore, we invite you to submit a revised version of the manuscript that addresses the points raised during the review process.

We would appreciate receiving your revised manuscript by Mar 21 2020 11:59PM. To enhance the reproducibility of your results, we recommend that if applicable you deposit your laboratory protocols in protocols.io, where a protocol can be assigned its own identifier (DOI) such that it can be cited independently in the future. For instructions see: http://journals.plos.org/plosone/s/submission-guidelines#loc-laboratory-protocols

We look forward to receiving your revised manuscript.

Kind regards,

Hoon Ryu, PhD

Academic Editor

PLOS ONE

2. We note that your study was approved by a number of ethics committees. Please ensure that you have included the full name of all the ethics committees that approved the various aspects of your study. This should be included in the (supplementary) Methods section.

"I have read the journal's policy and the authors of this manuscript have the following competing interests: T.M. and C.B. are founders, executive directors and shareholders of Oryzon Genomics S.A.; C.M., D.R., M.M.P.L., AO, J.S. and E.C. are employees; and A.E., N.G., N.V., M.F., M.M. and J.C.C.P. are former employees of Oryzon Genomics S.A.; I.F. is former member of the SAB of Oryzon Genomics S.A.  A.O., M.F., M.M., A.E., N.V, J.C.C.P., T.M., C.M., D.R., C.G.F, M.P., R.N., A.A., are listed as inventors on one or several of the following patent applications of Oryzon Genomics S.A. related to this work: WO2012/013728; WO2013/057320; WO2016/198649; WO2017/158136; WO2019/025588. The authors have no other relevant affiliations or financial involvement with any organization or entity with a financial interest in or financial conflict with the subject matter or materials discussed in the manuscript apart from those disclosed."

5. Please upload a copy of Supporting Information Tables S1 to S4 which you refer to in your text.

Reviewers' comments:

Reviewer's Responses to Questions

**Comments to the Author**

1. Is the manuscript technically sound, and do the data support the conclusions?

Reviewer #1: Yes

Reviewer #2: Yes

2. Has the statistical analysis been performed appropriately and rigorously? 

Reviewer #1: Yes

Reviewer #2: Yes

3. Have the authors made all data underlying the findings in their manuscript fully available?

Reviewer #1: Yes

Reviewer #2: Yes

4. Is the manuscript presented in an intelligible fashion and written in standard English?

Reviewer #1: Yes

Reviewer #2: Yes

5. Review Comments to the Author

Reviewer #1: In this impressive study, Maes et al. found a novel role of ORY-2001 for brain memory function in various Alzheimer’s disease animal models. Through a set of comprehensive in vitro and in vivo studies the authors have successfully proven the hypothesis that epigenetic modulation concerts neuronal plasticity, memory function, and behaviors. This paper is very strong scientifically. The authors have explored all the controls which prove the main theory. The amount of supporting data (including supplementary figures) is overwhelming. Last but not least, it is a nicely written and very well organized paper.

1. I have only a few minor comments about the effect of ORY-2001 for rescuing the memory deficit in SAMP8 mice. Generally, Morris water maze test is one of majorly well-known animal memory tests. In this regard, if the water maze test is added in figure 2 or 3, then the manuscript would be much stronger.

2. The conclusion part is not supportive enough and does not meet the level of this paper’s data quality. Accordingly, I recommend that the authors could rewrite the conclusion part adequately.

Anyways, despite the above critiques, I still believe this is a great paper that is worth being published in this journal.

Reviewer #2: The authors report the development of vafidemstat (ORY-2001), a brain penetrant inhibitor of KDM1A and MAOB. They show extensive in vitro and in vivo rodent data on its selectivity and effects on gene expression and behavior. They mine human neuropathology gene expression data to show that many of the target genes are increased in Alzheimer disease. Overall, they provide compelling data on the efficacy of this novel compound. However, the following items should be addressed:

1) Please rephrase lines 9-11 for clarity and define SAMP8 model.

2) The authors state that ORY-2001 does not cause sedation but do not provide clear data to support this. They reference Fig 8A and S4 Fig C which do not appear to provide any measures of sedation.

3) Lines 416 and 417 reference only the SAMP8 mice, yet it is stated that comparison to SAMR1 is made.

4) It is not clear what is meant by “regulated in LOAD vs normal samples” in line 485. This should be clarified.

5) More details about the NCBI GEO: GSE44770 dataset should be provided including the numbers in each group, how groups were defined (clinically or pathologically), and age differences in order to best interpret this data. The possibility that differences between AD and control are due to neuronal loss/atrophy should be addressed.

6) For Figure 6C, images should be labeled with the relevant stain.

7) Interactor ELISA needs to be better defined and clearly delineated in the Figure 11 legend.

6. PLOS authors have the option to publish the peer review history of their article (what does this mean?). If published, this will include your full peer review and any attached files.

Reviewer #1: No

Reviewer #2: No

---

## [Author Response · Author response to Decision Letter 0]

21 Apr 2020

RESPONSE TO THE EDITOR

We have uploaded the following documents into the editorial editor:

• Revised Manuscript with Track Changes 

• Manuscript 

• Fig1, Fig4 (substituted for quality)

• Fig 6, Fig11 (modified as requested by the reviewers) 

• Response to Reviewers. Note: we have no objection to the publication of the file; it also contains some additionally processed data. 

• S2 File; revised to include the full names of the ethical committees

• Supporting information Tables S1 to S5

All images were reviewed in PACE

The financial disclosure statement is updated as follows: 

None of the authors was a direct beneficiary but the studies were funded by Oryzon Genomics S.A. (https://www.oryzon.com) and co-funded by grants or loans to [1] Oryzon Genomics S.A., Cornellá de Llobregat, Spain or [2] Fundació Bosch i Gimpera, Universitat de Barcelona, Barcelona, Spain or [3] Institut de Neurociències, Universitat Autònoma de Barcelona, Bellaterra, Spain: [1] CEN-20081013 and CEN-20101023 of the CENIT program of CDTI, Spanish Ministry of Industry, Tourism and Commerce (https://www.cdti.es/); by grant [1] RD08-2-0014 of CIDEM, Generalitat de Catalunya (https://www.accio.gencat.cat/); by grants [1] 20100902VEN and 20150202 of the Alzheimer’s Drug Discovery Foundation (www.alzdiscovery.org); by [1] FP7 grant 278871 of the European Union (https://ec.europa.eu/research/fp7/index_en.cfm); and by [1, 2, 3] RTC-2015-3898-1 and [1, 2] RTC-2016-4955-1 of the RETOS program of CDTI, Spanish Ministry of Science, Innovation and Universities (https://www.cdti.es/).

We have uploaded “Chemoprobe-based KDM1A Interaction ELISA” on protocols.io. We have not obtained the doi yet: doing so makes the protocol un-editable and that would impede us to add the full citation for the current manuscript. The protocol can be viewed using the following link: 

https://www.protocols.io/private/6680F1E461F111EAA0250242AC110004.

Please instruct us on how to proceed.

1. We have reviewed the manuscript meets PLOS ONE's style requirements. 

2. We have reviewed the full names of the ethics committees; they are included in the S2 File.

3. With respect to the Competing Interests section I want to provide the following updated statement:

"I have read the journal's policy and the authors of this manuscript have the following competing interests: T.M. and C.B. are founders, executive directors and shareholders of Oryzon Genomics S.A.; C.M., D.R., M.M.P.L., AO, J.S. and E.C. are employees; and A.E., N.G., N.V., M.F., M.M. and J.C.C.P. are former employees of Oryzon Genomics S.A.; I.F. is former member of the SAB of Oryzon Genomics S.A. A.O., M.F., M.M., A.E., N.V, J.C.C.P., T.M., C.M., D.R., C.G.F, M.P., R.N., A.A., are listed as inventors on one or several of the following patent applications of Oryzon Genomics S.A. related to this work: WO2012/013728; WO2013/057320; WO2016/198649; WO2017/158136; WO2019/025588. The authors have no other relevant affiliations or financial involvement with any organization or entity with a financial interest in or financial conflict with the subject matter or materials discussed in the manuscript apart from those disclosed. This does not alter our adherence to PLOS ONE policies on sharing data and materials with the following exceptions: Polyphemous, proprietary software of Oryzon Genomics S.A., is not available as an exportable .exe code. 

4. We confirm we will provide repository information for our data at acceptance. The microarray data have been deposited at NCBI GEO. The other data will be provided on Mendeley, and included the reserved doi in the S2 File. The data is ready for uploading.

We have updated the Data deposition and software section in S2 File as follows: 

Original: 

Microarray data have been submitted to NCBI-GEO under accession number GSE100413. Confidential pre-publication access is available to the reviewers using the following token: clkpiywmjfwzjyj. Polyphemous is proprietary software of Oryzon Genomics S.A. and implemented inhouse as a hardware embedded microarray pipeline management tool. It is not available as an exportable .exe code.

Modified to:

The data corresponding to the figures in the manuscript have been deposited at Mendeley with doi:10.17632/sc5tfwt4nr.1. Microarray data have been submitted to NCBI-GEO under accession number GSE100413. Polyphemous is proprietary software of Oryzon Genomics S.A. and implemented inhouse as a hardware-embedded microarray pipeline management tool. It is not available as an exportable .exe code.

5. Supporting Information Tables S1 to S5 have been uploaded.

RESPONSE TO REVIEWERS

1. I have only a few minor comments about the effect of ORY-2001 for rescuing the memory deficit in SAMP8 mice. Generally, Morris water maze test is one of majorly well-known animal memory tests. In this regard, if the water maze test is added in figure 2 or 3, then the manuscript would be much stronger.

Indeed, the Morris Water Maze (MWM) is a standard test to measure memory deficits, and its use in SAMP8 mice has been described (Chen et al., 2004). We explored the MWM test in experiment 4. SAMR1 and SAMP8 groups (8 months old) learned to identify the location of the underwater platform during the training days. SAMP8 mice were slower than SAMR1 mice to reach the platform. 

There was a tendency, but no significant difference for SAMR1 to spend more time in the platform quadrant than SAMP8 mice on the test day; i.e. the test did not provide an a priori good window for testing. SAMP8 mice behaved very poorly: their “total distance travelled” was much reduced compared to the SAMR1 control (a highly significant difference); and they exhibited “passive coping” (i.e. floating) behaviour rather than the normal swimming activity shown by the SAMR1 control strain. The lack of a significant difference between SAMR1 and SAMP8, together with the lack motivation for exploration and floating behaviour of SAMP8 mice invalidated this test as a means to evaluate memory in these animals. 

On the contrary, SAMP8 and SAMR1 mice exhibited similar total exploration times in the NORT, allowing for a solid evaluation of the intended parameter. The differences in DI between SAMP8 and SAMR1 mice were highly significant and reproducible, which is why we opted for this particular test to explore different treatments, doses, duration, age, etc. 

2. The conclusion part is not supportive enough and does not meet the level of this paper’s data quality. Accordingly, I recommend that the authors could rewrite the conclusion part adequately. 

We didn’t want to be too redundant, but have now summarized the main conclusions in this section.

Anyways, despite the above critiques, I still believe this is a great paper that is worth being published in this journal.

Reviewer #2: The authors report the development of vafidemstat (ORY-2001), a brain penetrant inhibitor of KDM1A and MAOB. They show extensive in vitro and in vivo rodent data on its selectivity and effects on gene expression and behavior. They mine human neuropathology gene expression data to show that many of the target genes are increased in Alzheimer disease. Overall, they provide compelling data on the efficacy of this novel compound. However, the following items should be addressed:

1) Please rephrase lines 9-11 for clarity and define SAMP8 model.

Modified to: 

ORY-2001 efficiently inhibits brain KDM1A at doses suitable for long term treatment, and corrects memory deficit as assessed in the novel object recognition testing the Senescence Accelerated Mouse Prone 8 (SAMP8) model for accelerated aging and Alzheimer’s disease.

2) The authors state that ORY-2001 does not cause sedation but do not provide clear data to support this. They reference Fig 8A and S4 Fig C which do not appear to provide any measures of sedation.

Our statement on the absence of sedation by the compound derives from the previous evaluation of locomotor activity of these mice cohorts, as measured by the total distances travelled in open field test (Figure 7A), and also by assessment of the total number of entries in both arms of the EPM and rats (Figure 7C). Both parameters are standard measures of locomotor activity that are decreased after treatment with sedatives like Midazolam (Olson and Sherwin, 2006) or triazolam and lorazepam (Krsiak and Sulcova, 1990). Also, ORY-2001 did not have an anxiolytic effect in SAMP8 mice or in the rat isolation rearing model as measured by the time spent in the open arms of the EPM (Figure 7B and 7D). Therefore, we concluded there was no sedative effect. The lack of sedative effect is also exemplified in the S1 and S2 videos. 

3) Lines 416 and 417 reference only the SAMP8 mice, yet it is stated that comparison to SAMR1 is made. 

The phrase was modified to: 

“We performed a genome-wide microarray-based survey on pooled PFC samples from SAMR1 mice and vehicle and ORY-2001 treated SAMP8 mice from the “basal” cohort. All samples were compared to the vehicle treated SAMP8 mice, using two-color hybridization.”

4) It is not clear what is meant by “regulated in LOAD vs normal samples” in line 485. This should be clarified. 

Modified to “… are down-regulated in LOAD vs normal samples”. Some markers like S100A9 are clearly up-regulated, but we treated them earlier in the text. Many of the other makers identified are down-regulated, which could indeed reflect either a lack of response capacity or a loss of cells that normally express these genes, or both. 

5) More details about the NCBI GEO: GSE44770 dataset should be provided including the numbers in each group, how groups were defined (clinically or pathologically), and age differences in order to best interpret this data. The possibility that differences between AD and control are due to neuronal loss/atrophy should be addressed. 

The authors thank the reviewer for pointing this out, as indeed other potential variables should be considered as potential causes for differential gene expression between the “LOAD” and “control” groups. Age and gender are obvious ones to examine; and available in the public NCBI GEO: GSE44770 dataset. For better clarity, we have now included mean ages and numbers of the LOAD and AD groups in the figure legend.

According to the original article reporting the GSE44770 dataset, groups were defined both clinically and pathologically; subjects were diagnosed with LOAD at intake and each brain underwent extensive LOAD-related pathology examination. 

We analysed the age of the subjects included in the study and we identified an important difference in mean age between the LOAD (N = 129; average age = 82.1 years) and control (N = 101; average age = 65.2 years) groups. We also reviewed the male/female ratio in the dataset and found that it was 4.3 in the control group and 0.9 in the LOAD group, a substantial difference. These differences could be a source for differential gene expression (although on the other hand it is well known that gender and age are important risk factors for AD). 

To evaluate the relevance of the biomarkers with independence of age, we made a sub-selection of subjects, in a rather tight central age window (68 y < age < 75 y), in order to obtain LOAD and control subcohorts within the same age range (LOAD: N = 23; average age = 72.5 y; control: N = 21; average age = 71.4 y). 

Of course, this drastic reduction of the size of the population reduces the power of the analysis, and the signal was lost for 3 genes (CALB2, GNG4, NPAS4) that showed smaller sized differences in mean gene expression in the total population. However, the rest of the genes remained highly significant: S100A8 (FC = 4.24, p = 0.0002); S100A9 (FC = 3.93, p = 0.0001); EGR1 (FC = 0.54, p = 0.0022), EGR2 (FC = 0.57, p = 0.0028), FOS (FC = 1.71, p = 0.0044), PENK1 (FC = 0.63, p = 0.0037), RBM3 (FC = 0.43, p = 0.0004), GAD1 (FC = 0.42, p < 0.0001), DOC2A (FC = 0.84, p = 0.0204), UCHL1 (FC = 0.52, p = 0.0001). Such genes were confirmed to be differentially expressed, and all changes were in the same direction as the total sample population. 

Subsequently, in order to evaluate the relevance of the biomarkers with independence of gender, we evaluated males and females separately. Almost all markers remained highly significant and the direction and magnitude of change was similar in the female and male sub-cohorts, as can be seen in the images below (A - C). (NOTE images included in the Response to Reviewers pdf uploaded)

Therefore, unless the differences between LOAD and controls originate in yet another unidentified source of systematic variation, these genes appear to be robust markers reflecting genuine differences between LOAD and controls. 

Finally, with respect to the possibility that differences between AD and control are due to neuronal loss / atrophy; we agree this is clearly a possibility for genes specifically expressed in neurons, like UCHL1. 

6) For Figure 6C, images should be labelled with the relevant stain. 

Corrected.

7) Interactor ELISA needs to be better defined and clearly delineated in the Figure 11 legend.

We have incorporated a schematic drawing of (chemoprobe) interactor ELISAs in Figure 11 and adapted the legend. For the interactor ELISAs, the antibodies combinations used for capture and detection are listed in the S2 File.

---

## [Editor Report · Decision Letter 1]

6 May 2020

Modulation of KDM1A with vafidemstat rescues memory deficit and behavioral alterations.

PONE-D-20-00734R1

Dear Dr. Maes,

We are pleased to inform you that your manuscript has been judged scientifically suitable for publication and will be formally accepted for publication once it complies with all outstanding technical requirements.

With kind regards,

Hoon Ryu, PhD

Academic Editor

PLOS ONE

---

## [Editor Report · Acceptance letter]

15 May 2020

PONE-D-20-00734R1 

Modulation of KDM1A with vafidemstat rescues memory deficit and behavioral alterations. 

Dear Dr. Maes:

I am pleased to inform you that your manuscript has been deemed suitable for publication in PLOS ONE. Congratulations! Your manuscript is now with our production department. 

With kind regards,

on behalf of

Dr. Hoon Ryu 

Academic Editor

PLOS ONE